# Biases in the albedo sensitivity to deforestation in CMIP5 models and their impacts on the associated historical radiative forcing

Quentin Lejeune[1,2], Edouard L. Davin[2], Grégory Duveiller[3], Bas Crezee[2], Ronny Meier[2], Alessandro Cescatti[3], Sonia I. Seneviratne[2]

[1]Climate Analytics, Berlin, 10969, Germany
[2]Institute for Atmospheric and Climate Science, ETH Zurich, Zurich, 8092, Switzerland
[3]European Commission Joint Research Centre, Ispra (VA), 21027, Italy

*Correspondence to*: Quentin Lejeune (quentin.lejeune@climateanalytics.org)

**Abstract.** Climate model biases in the representation of albedo variations between land cover classes contribute to uncertainties on the climate impact of land cover changes since pre-industrial times, and especially on the associated radiative forcing. The recent publications of new observation-based datasets offer opportunities to investigate these biases and their impact on historical surface albedo changes in simulations from the fifth phase of the Coupled Model Intercomparison Project (CMIP5). Conducting such an assessment is however complicated by the non-availability of albedo values for specific land cover classes in CMIP, as well as the limited number of simulations isolating the land use forcing. In this study, we demonstrate the suitability of a new methodology to extract the albedo of trees and crops/grasses in standard climate model simulations. We then apply it to historical runs from 17 CMIP5 models and compare the obtained results to satellite-derived reference data. This allows us to identify substantial biases in the representation of the albedo of trees, crops/grasses, and the surface albedo change due to the transition between these two land cover classes in the analysed models. Additionally, we reconstruct the local surface albedo changes induced by historical conversions between trees and crops/grasses for 15 CMIP5 models. This allows us to derive estimates of the albedo-induced radiative forcing from land cover changes since pre-industrial times. We find a multi-model range from 0 to -0.17 W/m$^2$, with a mean value of -0.07 W/m$^2$. Constraining the surface albedo response to transitions between trees and crops/grasses from the models with satellite-derived data leads to a revised multi-model mean estimate of -0.09 W/m$^2$ but an increase in the multi-model range. However, after excluding one model with unrealistic conversion rates from trees to crops/grasses the remaining individual model results vary between -0.03 and -0.11 W/m$^2$. These numbers are at the lower end of the range provided by the IPCC AR5 (-0.15 +/- 0.10 W/m$^2$). The approach described in this study can be applied to other model simulations, such as those from CMIP6, especially as the evaluation diagnostic described here has been included in the ESMValTool v2.0.

# 1 Introduction

The landscape transformations imposed by anthropogenic activities have the potential to modify the climate (Foley *et al.*, 2005; Mahmood *et al.*, 2014). Since pre-industrial times, important Land Cover Changes (LCC) have predominantly led to the replacement of forests by shorter vegetation types such as crops and grasses over large inhabited areas (Ramankutty and Foley, 1999; Pongratz *et al.*, 2008; Hurtt *et al.*, 2011; Kaplan *et al.*, 2011). Associated alterations of land surface properties such as albedo, roughness and evaporative fraction have modified climate conditions through the so-called biogeophysical effects (Pongratz *et al.*, 2010; de Noblet-Ducoudré *et al.*, 2012; Lejeune, Seneviratne and Davin, 2017). The overall climate impact of the biogeophysical effects of historical LCC remains a matter of debate (Pitman *et al.*, 2009; de Noblet-Ducoudré *et al.*, 2012; Lejeune, Seneviratne and Davin, 2017; Duveiller *et al.*, 2018) due to uncertainties regarding the magnitude of the imposed land-cover perturbations (Schmidt *et al.*, 2012), the resulting alterations in land surface properties, the interplay between radiative (related to albedo) and non-radiative processes (related to changes in evaporative fraction and roughness), and the influence of atmospheric feedbacks and non-local effects (Winckler, Reick and Pongratz, 2017; Winckler *et al.*, 2019). Concerning the surface albedo more specifically, model studies concluded that historical LCC have led to large-scale increases in this variable (Betts *et al.*, 2007; Boisier *et al.*, 2013) because trees have a lower albedo than shorter vegetation types, especially in the presence of snow (Cescatti *et al.*, 2012; Li *et al.*, 2015). This has resulted in a cooling effect, and climate models have simulated an associated global Radiative Forcing (RF) close to -0.2 W/m$^2$ (Betts *et al.*, 2007; Davin, de Noblet-Ducoudré and Friedlingstein, 2007; Pongratz *et al.*, 2009). However, Myhre, Kvalevåg and Schaaf (2005) and Kvalevåg *et al.* (2010) have argued that climate models usually overestimate the albedo difference between natural vegetation and croplands in comparison to satellite-derived observational evidence. This is consistent with the weaker radiative forcing of -0.09 W/m$^2$ due to anthropogenic land cover change found by Myhre, Kvalevåg and Schaaf (2005), after combining a radiative transfer model with reconstitutions of past surface albedo changes based on satellite observations of the current vegetation land cover and its albedo, as well as a data set for potential natural vegetation. The Fifth Assessment Report (AR5) of the IPCC overall estimated that LCC since 1750 have rather led to a RF of -0.15 +/- 0.10 W/m$^2$ (G. Myhre *et al.*, 2013). A substantial spread in the surface albedo response to historical LCC has also been identified amongst the models participating in the LUCID project (de Noblet-Ducoudré *et al.*, 2012). The diversity of model parameterisations was estimated to be responsible for about half of it, while the remaining uncertainties result from differences in the magnitude of the prescribed land cover.

More recent model intercomparison efforts such as the fifth phase of the Coupled Model Intercomparison Project (CMIP5, Taylor, Stouffer and Meehl, 2012) offer new opportunities to assess the magnitude of these model disagreements, as well as our understanding of the impact of historical LCC on surface albedo and the associated RF. Nevertheless, such an investigation is complicated by the fact that the modelling groups participating in CMIP5 have not provided data on the albedo of specific land cover classes but only mean surface albedo values over grid cells, which often contain various land cover classes. Only a few modelling groups have conducted experiments to isolate the historical land use forcing. In parallel to recent model developments, studies giving insights from satellite data on the climate effect of LCC have been published (Li *et al.*, 2015;

Alkama and Cescatti, 2016; Duveiller, Hooker and Cescatti, 2018b). They provide high-resolution information on the potential changes in various surface variables in response to land-cover transitions, which constitutes a very good benchmark to evaluate how this aspect is represented in climate models. The analyses described in this study thus rely on both climate model runs and satellite-based observational datasets to pursue two main objectives: 1) the validation of a methodology to systematically evaluate the representation of the surface albedo difference resulting from conversions between the dominant land cover classes in climate models (i.e., trees and crops/grasses) in standard climate model runs (such as from CMIP), with the view to be integrated in the ESMValTool v2.0 (Eyring *et al.*, 2020), and 2) the assessment of the Radiative Forcing from historical LCC using historical CMIP5 model simulations as well as observations to constrain model biases.

This study is therefore divided in several parts. First, we present the employed methods and data. In particular, we introduce a new methodology to extract the surface albedo for two different land cover classes (trees or crops/grasses), or the potential surface albedo change caused by conversions between these land cover classes, in simulations for which climate variables are only available at the grid cell level (Section 2). Second, we evaluate how well this methodology performs by using climate model simulations that also provide sub-grid cell albedo values for specific underlying land-cover classes as a testbed (Section 3). Third, we apply this approach to CMIP5 simulations to extract the surface albedo where the underlying vegetation is either trees or crops/grasses, as well as the surface albedo change due to transitions between these land cover classes simulated for present-day conditions, and compare the obtained results to satellite-derived reference data (Section 4). Fourth, we reconstruct the surface albedo changes since preindustrial times in CMIP5 models, and calculate the associated RF (Section 5). We also discuss the spread in the obtained model results in light of the biases identified in Section 4, and apply an observational constraint based on satellite-derived evidence to refine our estimates of the RF from surface albedo changes induced by historical LCC. Eventually, we compare our findings to those of previous studies, and discuss their limitations as well as potential for follow-up analyses (Section 6).

## 2  Methods and Data

### 2.1 Observational data

#### 2.1.1   Albedo of land-cover classes

In this study we evaluate the monthly surface albedo simulated by climate models for two land-cover classes: crops/grasses (merged into one single land cover class) and trees, using reference estimates obtained from satellite measurements. It is important to note that in the analysed models as well as in satellite products the surface albedo is influenced by both the vegetation canopy and the soil reflectance, with the latter contribution being especially important in regions or periods where the leaf area index is low. For the sake of simplicity, in this study the formulation "albedo of a specific land cover class" is used while referring to this mixed contribution of the soil and canopy to the surface albedo.

The observed surface albedo for both trees and crops/grasses is derived using the 300 meter-resolution land cover information provided by GlobCover v2.3 (Arino *et al.*, 2012), collected between January 2005 and June 2006, in combination with the mean of the white-sky (bi-hemispherical) and black-sky (directional-hemispherical) shortwave albedo data at 0.05°-resolution from GlobAlbedo (Lewis *et al.*, 2012), available at monthly timescale for the 1998-2011 period. An optimal estimation

approach and a gap-filling technique based on the MODIS surface anisotropy dataset were used to integrate data derived from the Advanced Along-Track Scanning Radiometer (AATSR), SPOT4-VEGETATION, SPOT5-VEGETATION2, and MERIS instruments (Lewis *et al.*, 2013; Muller *et al.*, 2013). GlobAlbedo products generally showed good agreement with estimates from MODIS (global $R^2$ of 0.85) and were assessed to be of very good quality overall; problems associated with snow detection were identified but lead to most significant artifacts at very high latitudes (>70°, Muller *et al.*, 2013).

To extract the albedo from specific land-cover classes at a resolution of 2° (i.e., approximately equal to that of the model simulations), the GlobCover original data are first regridded from their original 300 m resolution to a regular 0.0025° grid. We also group some classes provided in the detailed classification from GlobCover into two broad land cover classes (trees and crops/grasses), which are comparable to those for which the land cover fraction was reported by CMIP5 modelling groups. Details on how this grouping was performed are provided in Table S1. Then, for each 0.05° grid cell of the GlobAlbedo dataset

which is occupied by at least 95% by either trees or crops/grasses according to the GlobCover product, the seasonal cycle of albedo for this specific land cover class is approximated from the monthly surface albedo climatology for this grid cell, computed over the full period covered by GlobAlbedo. The results are then aggregated at 2° resolution, i.e. for each 2° grid cell the albedo climatology of a specific land cover class is derived by calculating area-weighted averages over the 0.05° resolution grid cells it contains, and for which a land cover-specific seasonal cycle of albedo was previously identified.

Although we haven't considered in our analysis the so-called "mosaic" classes representative of heterogeneous landscapes in the GlobCover data, the employed 95% threshold means that up to 5% of each selected 0.05° grid cell may contain other land cover types than those belonging to the tree or crop/grass classes, thus potentially introducing a small error when retrieving their exact albedo values.

**2.1.2 Albedo changes associated with land-cover transitions**
The dataset of Duveiller, Hooker and Cescatti (2018a) – hereafter referred to as D18 – was used to evaluate the potential monthly surface albedo changes arising from land-cover transitions between trees and crops/grasses as simulated by CMIP5 models for present-day conditions. This 1° resolution observational dataset was derived by "unmixing" the monthly surface albedo climatology over the 2008-2012 period from collection v005 of the NASA's MODerate-resolution Imaging

Spectroradiometer (MODIS) MCD43C3 product (Schaaf *et al.*, 2002), using land cover information for the year 2010 from the ESA-CCI land-cover dataset (ESA Land Cover CCI Product User Guide Version 2. Tech. Rep., 2017, available at: http://maps.elie.ucl.ac.be/CCI/viewer/download/ESACCI-LC-PUG-v2.5.pdf). Their methodology is based on a "space-for-time" analogy, i.e. it assumes that surface albedo changes that would arise from a land cover transition from trees to crops/grasses, for example, can be approximated by spatial differences between albedo values of trees and crops/grasses over

neighbouring areas, assuming the two land cover classes experience a similar background climate. The albedo product that served as input to construct the D18 dataset had been filtered for quality using the provided quality flags; the underlying logic was to favour higher quality retrieval without excluding too many values.

We used a version of the D18 dataset that is based on a generic vegetation classification (IGBPgen) with only four land cover classes: trees, shrubs, crops/grasses and savannas. Imprecisions in land cover datasets are mostly confined to misclassifications between land cover types within these broad classes (e.g., between two types of trees) or the difficulty to properly identify medium-sized or mixed-type vegetation (i.e., shrub or savanna-like, see for example Bontemps *et al.*, 2011). In contrast, these products are best at distinguishing very distinct land cover classes such as trees and crops/grasses. Therefore, the satellite-derived albedo values of these two broad classes (retrieved following the methodology presented in Section 2.1.1) as well as their differences (obtained from the D18 data) are characterised by relatively low uncertainties. Cescatti *et al.* (2012) had overall identified a slight underestimation (by 0-0.03) of the MODIS albedo compared to in situ data from FLUXNET (Baldocchi *et al.*, 2001) for a dozen of crops/grasses sites in the northern mid-latitudes, but it is difficult to exactly quantify the biases of satellite-based surface albedo products as there does not exist a sufficiently extensive network of in situ measurements to serve as a benchmark.

For the part of the analysis in which we estimate the observation-constrained RF associated with historical LCC in CMIP5 simulations, we used an extended version of the dataset originally presented by D18 that has a broader spatial coverage in order to increase the spatial overlap between model and observational results. The product from D18 was gap-filled by training a random forest classifier to reproduce the data according to similarities in local climate, and then using the climate information to predict the surface albedo changes due to specific land-cover transitions where gaps existed in the data, following the methodological steps described by Duveiller *et al.* (2020). Some precautions were taken to ensure that these predicted outputs remain realistic. First, all areas in which neither of the two land cover classes involved in a given transition are present were removed. Second, the random forest is only used for interpolation, i.e. only using combinations of climate indicator values that are actually observed for the considered transition. Finally, a clear systematic bias of the classifier was corrected by applying a simple linear regression.

## 2.2   Climate model simulations from CMIP5

In this study, we reconstruct two different quantities in CMIP5 models: 1) the simulated present-day albedo of trees and crops/grasses, to evaluate the surface albedo change arising from a potential transition between these two classes against observational data, and 2) the historical surface albedo changes associated with transitions between trees and crops/grasses, followed by an assessment of their consequences in terms of Radiative Forcing.

The simulated monthly surface albedo climatology for trees and crops/grasses under present-day conditions is reconstructed from historical "all-forcings" simulations of 17 CMIP5 models (Taylor, Stouffer and Meehl, 2012) for which the required information on land cover, downwelling shortwave radiation, upwelling shortwave radiation and snow cover fraction is available (see Section 2.3.1). A list of these models as well as those included in further parts of the analysis is available in

Table S2. Land fractions covered by crops, grasses and pastures were provided separately by CMIP5 modelling groups, but were considered as one land cover class (crops/grasses) in this study to ensure consistency with the observational data from D18. In the specific case of the HadGEM2-ES model, the land cover fraction covered with "anthropogenic grass" (representative of agricultural land as this model does not simulate crops) was not available either via the grassFrac or cropFrac variables, but was reconstituted from another publicly available variable (landCoverFrac). Present-day surface albedo values and snow cover fractions are extracted from the last five-year period common to all models (i.e., 2000-2004), i.e. spanning a period similar in length and as close as possible to that covered by the albedo dataset used in D18. For four models (GFDL-CM3, GFDL-ESM2G, GFDL-ESM2M and HadGEM2-ES), the snow cover fraction outputs were not available but have been calculated from the snow mass values following the technique suggested and validated by Qu and Hall (2007): the snow cover fraction is assumed to be 1 at locations where the snow mass equals 60 kg/m$^2$, and to evolve as a linear function of snow mass where it equals between 0 and 60 kg/m$^2$. If several ensemble members differing only in terms of their initial conditions were available for one specific model, their ensemble mean was considered in the analysis.

We also reconstructed the surface albedo changes associated with historical transitions between trees and crops/grasses between pre-industrial conditions (equivalent to those of 1860 in CMIP5 and extracted from the first 200 years of the "piControl" experiments), and the 1981-2000 time period of historical "all-forcings" experiments. The reconstruction algorithm is applied to all CMIP5 models for which the required information on land cover, downwelling and upwelling shortwave radiation is available for at least two ensemble members of the analysed experiments (see Section 2.3.2 for a description of the reconstruction methodology). Since GFDL-ESM2G and GFDL-ESM2M are two very similar versions of the same model with only one ensemble member each, we have in this case treated them as ensemble members of the same model (referred to as GFDL-ESM2). In order to be able to compute the RF constrained by observations, the reconstructed historical surface albedo changes associated with transitions between trees and crops/grasses were regridded to 1°x1° resolution. We have focused on the transitions between trees and crops/grasses for consistency with the observational data of D18, but also assessed the sensitivity of our results when considering the historical impact of overall changes in tree cover (e.g., also including replacement of trees by shrubs or bare soil). Additionally, factorial experiments isolating the climate forcing of historical LCC were available for four models: CanESM2, CCSM4, GFDL-ESM2 and IPSL-CM5A-LR. These experiments constitute benchmarks to evaluate the reconstructed historical surface albedo changes; the validity of the reconstruction is thus discussed further below as well as in the Supplementary Material.

## 2.3 Principle of the reconstruction method

### 2.3.1 Reconstruction of the simulated present-day albedo of specific land-cover classes

The present-day albedo values from trees and crops and grasses ($\alpha_{tr}$ and $\alpha_{cg}$) in CMIP5 historical simulations are reconstructed using an "unmixing" method similar to those previously applied to satellite-derived observational data to extract the land surface characteristics of specific land cover types, including albedo (Li *et al.*, 2015; Chen and Dirmeyer, 2019), and notably

to obtain the data from D18 used as a reference for the evaluation of CMIP5 models in this study. We include information on the land fraction covered by shrubs in the methodology, but do not reconstruct the albedo of this land cover class ($\alpha_{sh}$) because

of its limited spatial occurrence.

Concretely, for every land grid cell $i$ we considered spatial windows of 25 grid cells (5 in both the latitudinal and longitudinal dimensions) and centred over $i$, hereafter referred to as "big boxes" (see Figure 1). Within each big box, for each month we thus have a sample of up to 25 values for surface albedo ($\alpha$) and the land cover fractions occupied by each of the three considered land cover classes ($lcf_{tr}$, $lcf_{sh}$, $lcf_{cg}$) over the same simulation period. Multi-linear regressions of $\alpha$ against $lcf_{tr}$,

$lcf_{sh}$ and $lcf_{cg}$ are then performed in order to obtain $\alpha_{tr}$, $\alpha_{sh}$ and $\alpha_{cg}$.

Variations in snow cover lead to large variations in surface albedo, therefore we focus on the identification of the albedo of trees and crops/grasses over grid cells with a snow cover fraction less than 0.1 (considered as snow-free), or greater than 0.9 (considered as snow-covered). For both albedo and snow cover fraction, we consider monthly climatological values for the 2000-2004 analysis period. In each big box and for a given month, if the grid cell at the centre $i$ is snow-free the regression is

conducted by considering only snow-free grid cells following:

$$\alpha^{sf} = \beta_0^{sf} + lcf_{tr} \times \beta_1^{sf} + lcf_{sh} \times \beta_2^{sf} + lcf_{cg} \times \beta_3^{sf} \ (1)$$

where $lcf_{tr}$, $lcf_{sh}$ and $lcf_{cg}$ are vectors containing up to 25 values, the $\beta$ coefficients are specific to each big box and each

month, and the superscript $sf$ stand for snow-free. Similarly, if $i$ is snow-covered the regression is conducted by considering only snow-covered grid cells:

$$\alpha^{sc} = \beta_0^{sc} + lcf_{tr} \times \beta_1^{sc} + lcf_{sh} \times \beta_2^{sc} + lcf_{cg} \times \beta_3^{sc} \ (2)$$

where the superscript $sc$ stands for snow-covered.

$\alpha_{tr}$ and $\alpha_{cg}$ over the central grid cell $i$ of the big box are eventually reconstructed by extrapolating the partial linear regression lines for cases where $lcf_{tr}$, $lcf_{sh}$ and $lcf_{cg}$ are equal to 100% following, in case $i$ is snow-free:

$$\alpha_{tr}^{sf}(i) = \beta_0^{sf} + \beta_1^{sf} \times 100\% \ (3)$$
$$\alpha_{cg}^{sf}(i) = \beta_0^{sf} + \beta_3^{sf} \times 100\% \ (4)$$

or, if $i$ is snow-covered:

$\alpha_{tr}^{sc}(i) = \beta_0^{sc} + \beta_1^{sc} \times 100\%$ (5)

$\alpha_{cg}^{sc}(i) = \beta_0^{sc} + \beta_3^{sc} \times 100\%$ (6)

This reconstruction method can only perform well over big boxes with sufficient land cover information. Therefore, each predictor ($lcf_{tr}$, $lcf_{sh}$, $lcf_{cg}$) is only included in the regression (i.e., its corresponding term is included in Equation (1) or (2))

if its value is greater than 0 in at least two snow-free (if $i$ is snow-free) or snow-covered grid cells (if $i$ is snow-covered). Moreover, the regressions are only conducted in the big boxes that have at least 15 grid cells (either snow-free or snow-covered) in which the sum of all the included predictors exceeds 90%. After this reconstruction few remaining albedo values which are physically impossible (i.e., either smaller than 0 or larger than 1) are filtered out. In a last step, grid cells for which the standard error of the regression is higher than 0.01, or where the land fraction covered by trees and crops/grasses is lower

than 20% are discarded.

The potential surface albedo change associated with a transition between trees and crops/grasses $\delta\alpha_{tr \rightarrow cg}$ is eventually calculated by looking at the difference between the reconstructed albedo of trees and crops/grasses, for each grid cell where both values were derived. As the fraction covered by trees and crops/grasses covary, the error associated with this difference strongly decreases compared to those of the albedo values of single land cover classes. The applied filtering criteria thus differ

in this case: We only discard grid cells for which the land fractions covered by trees or by crops/grasses are lower than 10% and where the standard error of the regression is higher than 0.001.

A diagnostic enabling the automated reconstruction of the albedo difference between trees and crops/grasses in CMIP simulations following the methodology described in this section has been implemented in the ESMValTool v2.0 (more details available in Eyring *et al.*, 2020).


### 2.3.2 Reconstruction of the simulated surface albedo changes due to historical deforestation

A similar approach based on local regression is used to reconstruct the simulated historical surface albedo changes associated with transitions between trees and crops/grasses that occurred between pre-industrial times and the 1981-2000 period

($\Delta\alpha_{tr \rightarrow cg}$). It has previously been used to derive local changes in temperature due historical LCC in CMIP5 simulations (Lejeune *et al.*, 2018). In this case, the spatial predictors used to explain historical surface albedo changes ($\Delta\alpha$) are the historical transition rate between trees and crops and grasses ($lcc_{tr \rightarrow cg}$), latitude (*lat*), longitude (*lon*), and elevation (*elev*), such that, for each month:

$\Delta\alpha = \gamma_0 + lcc_{tr \rightarrow cg} \times \gamma_1 + lat \times \gamma_2 + lon \times \gamma_3 + elev \times \gamma_4$ (7)

where $lcc_{tr \to cg}$, *lat*, *lon* and *elev* are vectors containing up to 25 values and the $\gamma$ coefficients are specific to each big box and each month. The regressions are conducted in big boxes containing at least 15 land grid cells to improve the quality of the reconstruction (Figure 1). The surface albedo change associated with historical, local transitions between trees and crops/grasses over the central grid cell $i$ of a big box is then obtained by scaling the results of this local regression with the historical conversion rate from trees to crops/grasses experienced over i (compared with pre-industrial conditions):

$$\Delta\alpha_{tr \to cg}(i) = lcc_{tr \to cg}(i) \times \gamma_1 \ (8)$$

An uncertainty range for $\Delta\alpha_{tr \to cg}$ is also computed by applying the regression to each ensemble simulation of a given model. Additionally, for each ensemble simulation and each big box, a jackknife resampling is conducted: As many times as there are land grid cells with non-missing values in the big box, an additional regression is computed after leaving out each time a different grid cell (Efron, 1982). The obtained estimates of $\Delta\alpha_{tr \to cg}$ thus amount to between 16 and 26 – depending on the number of land grid cells in the big box – multiplied by the number of ensemble members. We then retain the median of these estimates, which increases the robustness of our results by eliminating strong dependencies on single model grid cells.

## 2.4 Computation of the Radiative Forcing of historical conversions between trees and crops/grasses

The Radiative Forcing (RF), expressed here in W/m², is defined as the net change in the energy balance of the Earth system due to some imposed perturbation (G. Myhre *et al.*, 2013). In our case, this perturbation is a modification of surface albedo arising from land-cover changes, in particular transitions between trees and crops/grasses, which affects the amount of reflected shortwave radiation leaving the Earth system at the top of the atmosphere. By how much this amount changes depends on a so-called radiative kernel $K_{\alpha_s}$ (Soden *et al.*, 2008), defined in this case as the differential response in outgoing shortwave radiation at the top of the atmosphere to an incremental change in surface albedo $\delta\alpha_s$ (Bright and O'halloran, 2019):

$$RF = -K_{\alpha_s} \times \delta\alpha_s \ (9)$$

We employ here the monthly CERES-based albedo change kernel (CACK) v1.0. Based on a novel, simplified parameterisation of shortwave radiative transfer (Bright and O'halloran, 2019), it is driven with a 16-year (2001-2016) climatology of downwelling shortwave radiation values at the surface and the top of the atmosphere obtained from the Clouds and the Earth's Radiant Energy System (CERES) Energy Balance and Filled (EBAF) 1°-resolution products (CERES Science Team, 2018). CACK (hereafter also referred to as $K_{\alpha_s}^{CACK}$) is more easily understandable and easier to apply than kernels derived from climate models, while being able to mimic them more faithfully than five previously employed analytical, semi-empirical and empirical kernels (Bright and O'halloran, 2019). The reconstructed surface albedo changes caused by historical conversions

between trees and crops/grasses $\Delta\alpha_{tr\leftrightarrow cg}$ are also monthly, therefore the associated annual mean $RF_{tr\leftrightarrow cg}$ can be written as follows:

$$RF_{tr\leftrightarrow cg} = -\frac{1}{12}\sum_{m=1}^{12} K_{\alpha_s,m}^{CACK} \times \Delta\alpha_{tr\leftrightarrow cg,m} \quad (10)$$

where the subscript m denotes monthly values.

We derive two types of RF estimates in the analysed CMIP5 models. For the first one ("unconstrained"), which is purely model-based, we used the $\Delta\alpha_{tr\rightarrow cg}$ from historical conversion rates between trees and crops/grasses that were derived with the reconstruction method described in Section 2.3.2. The second one is constrained by observations, and was computed by combining the historical conversion rates implemented in the models $lcc_{tr\rightarrow cg}$ with the potential surface albedo change associated with a transition between trees and crops/grasses from D18 ($\delta\alpha_{tr\rightarrow cg}^{D18}$) such as:

$$\Delta\alpha_{tr\leftrightarrow cg,m}^{constrained} = lcc_{tr\rightarrow cg} \times \delta\alpha_{tr\rightarrow cg,m}^{D18} \quad (11)$$

and

$$RF_{tr\leftrightarrow cg}^{constrained} = -\frac{1}{12} \times lcc_{tr\rightarrow cg} \times \sum_{m=1}^{12} K_{\alpha_s,m}^{CACK} \times \delta\alpha_{tr\rightarrow cg,m}^{D18} \quad (12)$$

## 2.5   Additional simulations to evaluate the reconstruction method

We employ two additional offline simulations conducted with the Community Land Model version 4.5 (CLM4.5; Oleson *et al.*, 2013) to evaluate the ability of the reconstruction method presented in Section 2.3.1 to extract the simulated albedo of trees and crops/grasses. The simulations were conducted at 1.9°x2.5° resolution, forced by the CRUNCEP v4 atmospheric forcing dataset (Harris *et al.*, 2014) for the years 1997 to 2010, and we kept the 2002-2010 period from the analysis. The default land cover map of CLM4.5 was kept constant at the state of 2000 throughout the simulation period (Lawrence and Chase, 2007). Grid cells in CLM4.5 are divided into tiles of different land units (glacier, wetland, vegetated, lake, and urban); the vegetated land unit comprises tiles of different Plant Functional Types (PFTs) including several types of trees, shrubs, grasses and crops, which all receive the same atmospheric forcing. Surface albedo values were output for each tile in these simulations, enabling to extract a subgrid albedo value for each land cover class (trees or crops/grasses, similarly as in Malyshev *et al.*, 2015; Meier *et al.*, 2018). For each grid cell and each month, the albedo values for these two land cover classes are computed as the area weighted mean albedo across each PFT pertaining to the respective class over the analysis period. This reference value, later referred to as "subgrid" estimate, can then be compared to the albedo values obtained by applying the reconstruction method on these CLM4.5 simulations. The results of this evaluation are described in Section 3.

## 3 Evaluation of the methodology to reconstruct the simulated albedo of individual land cover classes

### 3.1 Reconstruction of the albedo of trees and crops/grasses

The reconstructed July albedo estimates for trees and crops/grasses are close to the subgrid reference values in the CLM4.5 simulations, for the grid cells where the reconstruction method yields results (Figure 2). The main patterns of the spatial variability of the albedo of both land cover classes of interest, such as their latitudinal variations, are captured by the reconstruction method. Globally the reconstructed and subgrid albedo estimates are highly correlated ($R^2$=0.91 for trees and 0.75 for crops/grasses). Differences between them indicate the "error" of the reconstruction, thus allowing to compute a global

Root Mean Square Error (RMSE) that considers all grid cells for which a reconstructed estimate could be derived. For the month of July, the global RMSE equals 0.0085 in the case of trees and 0.0097 for crops/grasses. Locally, the error is higher over some areas with stronger albedo gradients such as Western Europe, the Southeastern United States in the case of trees or Western Russia in the case of crops/grasses. Nevertheless, the absolute error rarely exceeds ~0.03, or ~20% of the subgrid values over these regions (Figure S1).

In January, the reconstructed albedo estimates still resemble closely the reference values from the subgrid model outputs (Figure 3). However, the presence of snow increases both the mean value and the spatial variability of albedo which results in higher RMSEs over grid cells located north of 40°N (0.037 for trees, respectively 0.0295 for crops/grasses as indicated in the right panel), leading to global RMSEs of 0.019, respectively 0.013. As a result, within one big box used for the reconstruction, the dispersion between the surface albedo values from individual grid cells is higher. This renders the extraction of the correct

albedo values of specific land cover classes with the regression-based reconstruction method more difficult. The spatial coverage of the reconstruction method also diminishes during months with a higher snow cover, because our methodology excludes grid cells which are neither considered snow-free nor snow-covered from the reconstruction, as is the case in Western Europe or the Northeastern United States in January. The absolute error of the reconstruction method reaches a maximum of ~0.1 or ~30-40% over localised parts of Eastern Siberia during this month (Figure S2).

### 3.2 Reconstruction of the surface albedo change arising from conversions between trees and crops/grasses

Overall, the reconstructed estimates of the July surface albedo change associated with a conversion between trees and crops/grasses also show a good correspondence with the subgrid reference values (Figure 4). The global RMSE increases up to 0.0189 in this case, because it is a combination of the errors from the reconstruction of the albedo of both trees and

crops/grasses. The magnitude of this error needs to be assessed in relation to the local albedo difference between trees and crops/grasses. Previous studies using satellite observations have shown that this difference roughly ranges between 0.03 and 0.07 over mid-latitudes during summer (Li *et al.*, 2015; Duveiller, Hooker and Cescatti, 2018b). This means that in most regions the local difference between the reconstructed and subgrid estimates remains less than the albedo difference between trees and crops/grasses, but can attain similar magnitudes in some regions such as Western Europe or the Northeastern United

States (Figure S3).

For January, the reconstructed and subgrid estimates of the deforestation-induced surface albedo change remain similar to each other (Figure 5), with a global RMSE that slightly increases to 0.025 and reaches 0.0505 on average north of 40°N. The relative error between the reconstructed and subgrid albedo values is similar as in July over localised tropical or subtropical areas where it can reach 80%, whereas it mostly remains limited to +/-10% over snow-covered regions (Figure S4). This is because the absolute error remains of similar magnitude as in July in snow-free regions, while the surface albedo change induced by deforestation increases in the presence of snow due to the snow-masking effect of trees.

Overall, the reconstruction method yields similar estimates of the absolute albedo of trees and crops/grasses (Section 3.1), and a similar albedo difference between these two land cover classes as the subgrid reference values in the analysed CLM4.5 simulations. It is nevertheless associated with an error that varies with the season and more particularly with the presence of snow. These uncertainties introduced through the reconstruction method need to be kept in mind in the upcoming section, where the reconstruction method is applied to CMIP5 simulations and the resulting albedo estimates of trees, crops/grasses as well as the difference between the two are compared to satellite-derived reference values.

## 4 Present-day potential surface albedo changes associated with a transition from trees to crops/grasses in CMIP5 models and observations

### 4.1 Evaluation of the present-day albedo of trees and crops/grasses in CMIP5 models

#### 4.1.1 Albedo of trees

The reconstructed albedo of trees varies considerably across the analysed CMIP5 models for the month of July, especially over the mid-to-high latitudes (Figure 6). Estimates derived from CanESM2, HadGEM2-ES and the models from the MPI suite (MPI-ESM-LR, MPI-ESM-MR, MPI-ESM-P) show the highest similarities with the observed ones over regions where results from our reconstructions and observation-based data can be compared. The climate models which use the CLM as a land surface scheme (CCSM4, CESM1-CAM5, CESM1-FASTCHEM, CESM1-WACCM, NorESM1-M, NorESM1-ME) as well as MIROC5 all underestimate the albedo of trees over mid-to-high latitudes. They indeed simulate values lower than 0.1, whereas the estimates derived from observational data always remain above 0.1, and mostly range between 0.12 and 0.16 over these regions. On the other hand, the models from the GFDL suite (GFDL-CM3, GFDL-ESM2G, GFDL-ESM2M) exhibit higher tree albedo values than the in the reference data, especially over tropical regions where the overestimation can be as high as ~0.1. Lastly, our results indicate strong spatial variations in the case of the MIROC-ESM and MIROC-ESM-CHEM models, with negative biases over the high latitudes and Southeast Asia. The magnitude of these differences between reconstructed and reference estimates is significantly higher than the reconstruction error which has been assessed from the analysis of the CLM4.5 simulations (global RMSE of 0.0085, see Section 3.1).

For January, surface albedo increases over the regions where snow is present are reflected in the reference data (Figure 7). A latitudinal gradient can especially be noted, as the values derived from GlobCover and GlobAlbedo typically barely exceed

0.15 in Western Europe, but are higher than 0.3 in Scandinavia and even reach ~0.5 in Northern Siberia. Our results show that CanESM2 and the climate models using the CLM also simulate higher tree albedo values over snow-covered regions, with values that remain within the range indicated by observations for this time of the year. However, the models from the GFDL suite and especially GFDL-CM3 present an overestimation of these quantities, a behaviour that is even more pronounced in MIROC5 which exhibits values exceeding 0.5 north of ~50°N, and even reaching ~0.7 in areas located close to the Arctic

ocean. Such biases have already been reported for GFDL-ESM2M and MIROC5 and linked to unrealistic parameterisations of snow canopy and vegetation masking (Thackeray, Fletcher and Derksen, 2015). They are significantly higher than the typical error of the reconstruction method identified for this month north of 40°N (~0.037). Unfortunately, in the case of HadGEM2-ES, MIROC-ESM, MIROC-ESM-CHEM and the models from the MPI suite (MPI-ESM-LR, MPI-ESM-MR and MPI-ESM-P) the spatial coverage of the reconstruction method is too low to be able to draw meaningful comparisons with

observations over snow-covered areas.

### 4.1.2    Albedo of crops/grasses

There are also important variations among the simulated albedo of crops and grasses in the CMIP5 models we have analysed,

pointing to significant model biases in comparison to observation-derived reference estimates. Overall, the models that employ the CLM tend to underestimate this quantity over large parts of the tropics and the mid-latitudes in the Northern Hemisphere in July, with reconstructed albedo values of ~0.13-0.14 whereas observations rather indicate values of at least 0.15 and even approaching 0.25 over the Sahel and Central Asia (Figure 8). This discrepancy appears less pronounced over the tropical parts of Africa and America located in the Southern Hemisphere, despite the lower availability of observational estimates over these

regions. Our results also reveal that MIROC5 more systematically underestimates the albedo of crops/grasses, which remain less than 0.15 worldwide in this model. In contrast, the models from the MPI suite simulate albedo values that are consistently greater than those of the observations, exceeding 0.2 over large regions of the world. These overestimations are often higher in the GFDL models, especially over Central Asia, the southern part of South America or the southern tip of Africa, although these three models present an opposite behaviour over equatorial regions of America and Africa with remarkably low albedo

values. Importantly, these numerous reported differences between the reconstructed model estimates and the reference values from observations are significantly higher than the error of the reconstruction method derived from the analysis of the CLM4.5 simulations (~0.01 in the case of crops/grasses for the month of July, see Section 3.1). The albedo values simulated by HadGEM2-ES, MIROC-ESM and MIROC-ESM-CHEM appear closer to the observational estimates over the regions where those are available. Lastly, the spatial coverage of the reconstruction is low in the case of CanESM2, which prevents drawing

robust conclusions for this model.

Results for the month of January indicate that the models that include the CLM, as well as MIROC5 and those from the MPI and GFDL suites all represent the increase in the albedo of crops/grasses over snow-covered areas which is indicated by observational estimates (Figure 9). The limited spatial coverage of the latter over the high latitudes however makes it difficult

to evaluate whether the magnitude of this increase is correctly represented. Over the tropical regions, the models including the CLM simulate an opposite pattern compared to that shown for the month of July, i.e. an underestimation of the albedo of crops/grasses in the Southern Hemisphere but more realistic estimates in the Northern Hemisphere. This suggests that these models simulate too high variations of the annual cycle for this variable over tropical regions.

## 4.2   Evaluation of the surface albedo changes induced by a transition from trees to crops/grasses in CMIP5 models

The observational dataset from D18 indicates that the conversion of trees into crops/grasses leads to a higher local albedo over each region of the world it covers, with some spatial variations in the magnitude of this increase. In July, this increase is lowest (<0.01) over Eastern Asia and Southwestern Siberia, and highest (~0.1) over the western part of North America (Figure 10). Our reconstructions indicate that most of the analysed CMIP5 models simulate the albedo increase induced by the replacement of trees by crops/grasses over most regions of the world. However, there are biases that are significantly higher than the typical error of the reconstruction method derived from its evaluation on CLM4.5 simulations (~0.02 in July, see Section 3.2). At this time of the year, the CanESM2, HadGEM2-ES and MIROC5 models show the closest resemblance to the observational reference data, nevertheless they overestimate the albedo increase due to the conversion of trees into crops/grasses over some regions such as Eastern Asia and the reconstructed results are available over limited areas for these models. As a result of their strong overestimation of the albedo of crops/grasses (see Section 4.1), the models from the MPI suite exhibit significant positive biases in the deforestation-induced albedo increases across the globe in July, with values reaching ~0.1 over large areas. Positive biases of a lower magnitude, although still significant, are also found over specific regions in the models using the CLM as a land surface scheme, consistently with the evaluation of the subgrid albedo difference in CLM4.5 by Meier *et al.* (2018). Over the mid-latitudes, this is due to the underestimation in the albedo of trees, whereas it can be related to the too high albedo of crops/grasses over the tropical regions of the Southern Hemisphere for this time of the year. Lastly, the MIROC-ESM, MIROC-ESM-CHEM and GFDL models exhibit a strong spatial variability in the reconstructed signals. In contrast with the observational data which consistently indicate an increase in albedo after conversions of trees into crops/grasses, our estimates suggest that the former two simulate the opposite behaviour over extensive areas of Central Asia, but also the western parts of Canada and the United States and south of 25°S in Africa, America and Western Australia. As for the GFDL models, similarly to the models from the MPI suite they exhibit an overestimation of the albedo of crops/grasses in July (Section 4.1.2), but also an overestimation of the albedo of trees in many regions and both tend to compensate in some regions. This leads to the described spatial variability in the biases associated with the deforestation-induced albedo increase, which can even become negative over Europe although their limited magnitude suggests to interpret them with caution, in light of the error of the reconstruction method. The negative biases over the equatorial band can however directly be linked to the very low albedo of crops/grasses reported in these regions and for these models (Section 4.2.2).

Compared to July, the observations of D18 for the month of January indicate a higher albedo increase following conversion of trees into crops/grasses over the mid-to-high latitudes where snow is present, the magnitude of which is overestimated by ~0.05-0.1 by the CMIP5 models including the CLM (Figure 11). This is slightly higher than the typical error of the

reconstruction method (~0.05 north of 40°N), and in line with the findings of Meier *et al.*, 2018. These models also consistently simulate a localised, likely non-significant albedo decrease following the replacement of trees by crops/grasses over Eastern
Europe, a feature that is not present in the observations. Strikingly, our results suggest that the MIROC-ESM and MIROC-ESM-CHEM models simulate strong albedo decreases (below -0.3) over large-snow covered regions at this time of the year, a behaviour that is in strong contradiction with what observational data indicate. In line with the overestimation of the albedo of trees over high latitudes represented by MIROC5, this model also simulates albedo decreases as a response to trees replacement by crops/grasses over parts of Europe.


## 5    Implications for the Radiative Forcing from historical deforestation

Our reconstructions of the RF from transitions between trees and crops/grasses since preindustrial times indicate a large spread within the CMIP5 models which were considered in this analysis (Figure 12), with estimates of the global mean RF ranging
between 0 and -0.17 W/m². This dispersion is due to differences in two factors across the models: their local surface albedo responses to a transition between trees and crops/grasses, and the historical conversion rates between these two land cover classes that the models simulate or prescribe (depending on whether they used a dynamic vegetation module or not). In Eq. (8), the former factor is represented by $\gamma_1$, and the latter by $lcc_{tr \to cg}$. Observation-constrained estimates of the RF from the historical conversion rates in CMIP5 models were obtained by replacing the reconstructed values of $\gamma_1$ by those from D18
(Figure 13, see also Sections 2.3.2 and 2.4 for more information on the methodology). The differences between the unconstrained and constrained RF values therefore reflect the model biases in the local surface albedo response to a present-day conversion from trees to crops/grasses, which have been described in Section 4.2 for a subset of the models considered here for the months of July and January. Hence, the constrained global RF estimates from the models using the CLM as a land surface scheme (CCSM4, CESM1-CAM5, CESM1-FASTCHEM, NorESM1-M), as well as those from the MPI suite (MPI-
ESM-LR, MPI-ESM-MR, MPI-ESM-P) are less negative than the unconstrained estimates by 0.01-0.02, respectively 0.04-0.07 W/m², reflecting the fact that these models were found to overestimate the surface albedo increase via this land cover transition. On the other hand, the low surface albedo response exhibited by MIROC5 in snow-covered regions can be related to the slightly more negative RF (by 0.01W/m²) obtained for this model after constraining it with the observational data from D18. Similarly, the mix of surface albedo decreases and increases following a present-day transition from trees to crops/grasses
that have been identified for MIROC-ESM both in January and July can also explain that the global reconstructed RF equals zero for this model, whereas it reaches -0.23W/m² after applying the same observational constraint. As for the GFDL-CM3 and GFDL-ESM2 models, the unconstrained global RF values become more negative by 0.05-0.06 W/m² once constrained with the observations from D18, reflecting the locally low or negative surface albedo sensitivity to conversions of trees into crops/grasses described in Section 4.2 but also suggesting other important biases at very high latitudes, where the reconstructed
model estimates could not be derived. The constrained global RF is more negative (respectively, less negative) than the unconstrained estimate by 0.03 W/m² (respectively, 0.02 W/m²) in the case of CanESM2 (respectively, HadGEM2-ES). We

have previously identified close resemblance with observations but limited spatial coverage of the reconstructed results for these two models, however this additional RF analysis suggests that they underestimate (in the case of CanESM2) or slightly overestimate (in the case of HadGEM2-ES) the deforestation-induced surface albedo change over areas where this response

could not be reconstructed. Lastly, the constrained and unconstrained estimates of the IPSL-CM5A-LR and IPSL-CM5A-MR models are very similar, suggesting that the surface albedo response to a conversion between trees and crops/grasses simulated by these models is close to the observed values.

Although it solely reflects the model spread in the historical conversion rates between trees and crops/grasses $lcc_{tr \rightarrow cg}$, the dispersion between the constrained estimates of the global RF is higher than between the unconstrained ones (Figure 14). This

is due to the MIROC-ESM model in particular, for which the $lcc_{tr \rightarrow cg}$ value constitutes an outlier among the whole set of models, but which at the same time exhibits significant biases in its surface albedo response to these LCC. Thus, the unconstrained RF equals zero for this model, which is in line with a globally averaged surface albedo response to transitions between trees and crops/grasses that is also equal to zero, as described above in this Section. In contrast, MIROC-ESM also exhibits the strongest constrained estimate (with -0.23 W/m$^2$) because of the strong historical conversion rates it simulates,

which exceed 50% over large areas of Australia, North America, southeastern Brazil, Central Asia and southern Africa (Figure S5).

The extremely high historical conversion rates from trees to crops/grasses in MIROC-ESM cast doubt on the global RF obtained for this model. In Figure 14 we therefore also show the model spread after omission of the results of this model, which is reduced from 0.16 to 0.08 W/m$^2$ after applying the observational constraint. Constraining based on the D18 data also

leads to a slightly more negative model mean value (-0.09 W/m$^2$ instead of -0.07, note that the models including the same land surface scheme and land cover maps are considered as just one model for the computation of the mean).

For most CMIP5 models, our reconstructions indicate that the historical impact of conversions between trees and crops/grasses on surface albedo is very similar to that arising from all changes in tree cover (i.e., also including for example the replacement of trees by shrubs and bare soil, or vice-versa, see Figures S6-20). Moreover, we also find a similar effect for surface albedo

variations from all LCC (i.e., also including transitions between shrubs, crops/grasses and bare soil) by comparing experiments with and without the land-cover forcing, available for four of the analysed models (CanESM2, CCSM4, GFDL-ESM2 and IPSL-CM5A-LR, see Figures S6, S7, S11 and S13). Since it solely considers the transition between trees and crops/grasses, this method likely also slightly overestimates the RF for MPI-ESM-LR, MPI-ESM-MR and MPI-ESM-P (Figures S17-S19), because these three models represent an expansion of both trees and crops/grasses over high latitudes. Despite these limitations,

our analysis shows that the reconstructed RF from historical transitions between trees and crops/grasses are overall good approximations of the RF from all LCC for most of the analysed CMIP5 models (see also Figure S21).

## 6    Discussion and conclusions

The conclusions that can be drawn from the presented analysis are manifold. First, we introduced a methodology to derive the albedo of trees and crops/grasses from Earth System model simulations that only provide mean surface albedo values over grid cells containing a mix of land cover classes. This "reconstruction" method employs multi-linear regressions to disentangle local information on land cover and surface albedo within moving windows ("big boxes") encompassing several grid cells. It assumes that spatial albedo variations between neighbouring trees and crops/grasses within a big box are good proxies of the

potential surface albedo change arising from a transition between these two land cover classes. We then demonstrated that in the Community Land Model the estimated albedos of trees and crops/grasses from the reconstruction method are close to the values provided at the sub-gridcell level. Consequently, as a second step we reconstructed the present-day albedo of trees and crops/grasses in CMIP5 simulations for 17 models, and compared the obtained results with reference values from observations. Despite the relatively low spatial coverage of the reconstructed estimates in some models, especially over regions where snow

is present, we were able to identify substantial model biases for the months of January and July which are significantly higher than the error of the reconstruction method. We found that they are reflected further in the representation of the surface albedo change induced by a transition between trees and crops/grasses in the same CMIP5 models. Finally, we investigated how such model biases influence the surface albedo change due to historical transitions between trees and crops/grasses as simulated by CMIP5 models, as well as the associated Radiative Forcing. To do so, we used another reconstruction methodology, already

employed in previous studies, to assess how surface albedo has been modified as a result of the replacement of trees by crops/grasses since pre-industrial times in 15 CMIP5 models (including most of those analysed in the previous step). We then derived the associated historical RF by using a recently published kernel, which constitutes of a simple parameterisation found to mimic the behaviour of climate models and applied to CERES radiation observations. An observational constraint was also applied to these estimates, by replacing the reconstructed surface albedo response to a conversion from trees to

crops/grasses in the models by that of the observational dataset previously used for the model evaluation. The comparison of the unconstrained and observation-constrained RF in the individual models revealed differences reflecting some of the model biases that we had previously described. Moreover, the observational constraint leads to a multi-model mean RF associated with the historical replacement of trees by crops/grasses that is slightly reduced from -0.07 to -0.09 W/m$^2$, and a model range spanning from -0.03 to -0.11 W/m$^2$ after excluding one model outlier with unrealistically high historical conversion rates

between trees and crops/grasses. Considering all variations in tree cover or even all Land Cover Changes gives very similar results, because of the simplified representation of land cover in CMIP5 models.

Our RF estimates were derived with all-forcing simulations in which climate is evolving mostly due to other forcings (G. Myhre *et al.*, 2013), and thus are theoretically not exactly comparable with results from studies that assessed the impact of historical LCC in isolation from other forcings. However, our finding that the reconstructed albedo values are similar to those

derived with LCC-only experiments conducted within CMIP5 (see Figures S6 and S7) indicate that changes in background climate from other forcings have had little influence on the overall LCC-induced surface albedo changes over the 1860-2000 period. This confirms earlier similar conclusions (Boisier *et al.*, 2012; de Noblet-Ducoudré *et al.*, 2012) and also suggests that the reconstructed RF values are similar as well to those that can be calculated based on LCC-only experiments (Figure S21),

and are thus comparable to estimates from previous model-based studies. The identified range of -0.03 to -0.11 W/m$^2$ for the global RF is at the lower end of that of -0.15 +/- 0.10 W/m$^2$ provided by the IPCC AR5 (i.e., less negative than its best estimate, see Figure 14). This result confirms that the LCC forcing is unlikely to have played a large role historically for global mean impacts (G. Myhre *et al.*, 2013; Smith *et al.*, 2020), while still being important at the local to regional scales (Pongratz *et al.*, 2010; Boisier *et al.*, 2012; de Noblet-Ducoudré *et al.*, 2012). It is also lower than the estimates close to -0.2 W/m$^2$ from Betts *et al.*, (2007); Davin, de Noblet-Ducoudré and Friedlingstein (2007) and Pongratz *et al.* (2009). Myhre, Kvalevåg and Schaaf (2005) and Kvalevåg *et al.* (2010) suggested that these climate model-based studies had overestimated the simulated surface albedo response to historical LCC. In this regard, our study reveals that such an overestimation does exist for some CMIP5 models, but is not systematic across the analysed ensemble. Our model mean result is equal to, respectively lower than those of Myhre, Kvalevåg and Schaaf (2005) and Ghimire *et al.* (2014), who both used satellite data to reconstruct past surface albedo changes and found RFs of -0.09 and -0.15 W/m$^2$ when considering all LCC since pre-agricultural times. It is also slightly less negative than – although within the uncertainty range – of the multi-model mean RF of -0.14 W/m$^2$ estimated within the Radiative Forcing Model Intercomparison Project (RFMIP) as part of the 6$^{th}$ phase of CMIP (CMIP6, Smith *et al.*, 2020), which also found that it would translate into an Effective Radiative Forcing of -0.09 W/m$^2$ after adjustment of the state of the troposphere (clouds, water vapour content, etc.).

Additionally, part of these differences and part of the model spread identified in this study arise from different simulated historical conversion rates from trees to crops/grasses. Despite being based on the same Land Use History a product (LUHa, Hurtt *et al.*, 2011), the LCC trajectories in the analysed CMIP5 historical simulations reflect varying interpretations of this dataset. LUHa gives gridded information on annual transitions between four types of land use (primary land, secondary land, crop and pasture) for the 1500-2005 period, which were derived with the Global Land use Model (GLM) based on historical data. These transitions were specifically designed to provide common reference land use trajectories for all historical CMIP5 simulations. The CMIP5 models may have however considered that primary and secondary land were covered by either trees or crops/grasses, or even shrubs or bare soil, depending on the land cover distributions that were prescribed or simulated in a given region or under a given climate. These different interpretations of common land use input data contribute substantially to the spread in the surface albedo variations due to historical LCC. This result had already been identified for the models participating to the LUCID project (Boisier *et al.*, 2012), as well as more generally for the biogeophysical effect of future LCC on climate in RCP4.5 and RCP8.5 simulations from CMIP5 (Brovkin *et al.*, 2013; Davies-Barnard *et al.*, 2014; Di Vittorio *et al.*, 2014). Solutions have been put forward to reduce the room for interpretation of the imposed land cover forcing in future model intercomparison efforts, such as a direct coupling between the Integrated Assessment Models producing the land cover scenarios and the Earth System Models (Di Vittorio *et al.*, 2014), or the provision of more detailed land cover information (including the land cover fractions allocated to several specific land-use states) in the frame of CMIP6 (Lawrence *et al.*, 2016). These may bring the multi-model mean RF estimate of LCC-induced historical albedo changes closer to the -0.15 W/m$^2$ put forward by Ghimire *et al.*, (2014), who combined the LUHa product with an observational constraint based on satellite data. The fact that Smith *et al.* (2020) found a slightly more negative multi-model mean RF (-0.14 W/m$^2$) than our best estimate

using RFMIP experiments may suggest it, but further analysis of CMIP6 simulations and notably within the Land Use Model Intercomparison Project (LUMIP, Lawrence *et al.*, 2016) are needed before robust conclusions can be drawn. Furthermore, there also exist uncertainties about the HYDE3.1 dataset (Klein Goldewijk *et al.*, 2011) on which the LUHa product is based, as significant differences with the land cover reconstructions from Kaplan *et al.* (2011) and Pongratz *et al.* (2008) have been identified (Schmidt *et al.*, 2012).

The analysis of the biases in the representation of the albedo of trees and crops/grasses in CMIP5 models performed in this study focussed on the months of January and July, during which the snow cover fraction is rather correctly represented in CMIP5 models (Thackeray, Fletcher and Derksen, 2015). It could however be repeated for other months and especially in spring and autumn, where misrepresentations of the timing of snow accumulation and melt as well as snow aging processes, leaf area index parameterisations and the ensuing vegetation masking effect on snow have been identified over the boreal latitudes (Thackeray, Fletcher and Derksen, 2014, 2015; Wang *et al.*, 2016).

When interpreting the findings presented in this study, it also needs to be kept in mind that the RF framework is not sufficient to capture the impact of LCC on other climate variables than albedo, as it cannot represent their non-radiative biogeophysical effects (i.e., that solely affect the partitioning between latent and sensible heat fluxes, see e.g. Davin, de Noblet-Ducoudré and Friedlingstein, 2007). Moreover, in this study we have focused our attention on local LCC-induced surface albedo changes, although those also led to an important remote cooling in global-scale deforestation experiments conducted with the IPSL model (Davin and de Noblet-Ducoudré, 2010).

In conclusion, we demonstrated the suitability of a new methodology to extract the albedo of trees and crops/grasses in ESM simulations that only provide mean surface albedo values over grid cells containing a diversity of land cover classes. After applying it to historical CMIP5 simulations, we identified significant model biases in the representation of the albedo of both trees and crops/grasses, as well as the surface albedo change arising from a transition between these two land cover classes. Additionally, we reconstructed local surface albedo modifications due to historical LCC. Since these reconstructions are affected by model biases, we used the observed surface albedo response to transitions between trees and crops/grasses to derive an observation-constrained RF of historical LCC in CMIP5 models. Compared to IPCC AR5 estimates, our results point to a slightly less strong global mean RF, with some remaining uncertainty due to the various magnitudes of LCC implemented in the analysed models. With the release of new ESM simulations within CMIP6 (Eyring *et al.*, 2016), new opportunities arise to assess whether the biases identified in this study have been corrected in the latest generation of ESMs. In that respect, the reconstruction methodology developed for this analysis and which has been implemented as a diagnostic in the ESMValTool v2.0 (Eyring *et al.*, 2020) should allow for a more straightforward model evaluation. Additionally, the new approach to harmonise the forcing from historical LCC in CMIP6 may enable to identify a refined estimate of their RF. We advance that combining recently released observational evidence and model results as proposed in this study will be useful in this context, in order to further reduce uncertainties on the climate impact of historical LCC on both global and local scales.

## 7    Code availability

The code to reproduce the results and figures presented in this study can be made available upon request.

## 8    Supplement link


## 9    Author contribution

Q.L., E.L.D. and S.I.S. designed the study. Q.L. wrote the code to conduct the analysis, with some contributions from B.C. G.D. produced new satellite-based data to evaluate the model results. R.M. conducted the offline simulations with CLM4.5. Q.L., E.L.D., G.D., A.C. and S.I.S. interpreted the results. Q.L. prepared the manuscript with contributions from E.L.D., G.D.,

B.C. and R.M.

## 10    Competing interests

The authors declare that they have no conflict of interest.

## 11    Acknowledgements


We acknowledge the World Climate Research Programme's Working Group on Coupled Modelling, which is responsible for CMIP, and we thank the climate modelling groups for producing and making available their model output. For CMIP the U.S. Department of Energy's Program for Climate Model Diagnosis and Intercomparison provides coordinating support and led development of software infrastructure in partnership with the Global Organization for Earth System Science Portals. We

thank V. Avora, I. Bethke and D. Lawrence for providing additional, not publicly available data from CMIP5 simulations. We are also very grateful to U. Beyerle and R. Wartenburger for managing the CMIP5 and Earth observation databases at ETH. Q.L., E.L.D., B.C. and S.I.S acknowledge funding from the European Union's Horizon 2020 Framework Programme for Research and Innovation CRESCENDO project (https://www.crescendoproject.eu/; grant no. 641816). Q.L. also received funding from the German Federal Ministry of Education and Research (BMBF) and the German Aerospace Center (DLR) via

the LAMACLIMA project as part of AXIS, an ERA-NET initiated by JPI Climate (http://www.jpi-climate.eu/AXIS/Activities/LAMACLIMA, grant no. 01LS1905A), with co-funding from the European Union (grant no. 776608). E.L.D and R.M acknowledge funding from the Swiss National Science Foundation (SNSF) and the Swiss Federal Office for the Environment (FOEN) through the CLIMPULSE project (http://p3.snf.ch/Project-172715; grant no. 200021_172715). S.I.S. acknowledges partial funding from the European Research Council (ERC) 'DROUGHT-HEAT'

project funded by the European Community's Seventh Framework Programme (grant agreement FP7-IDEAS-ERC-617518).

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

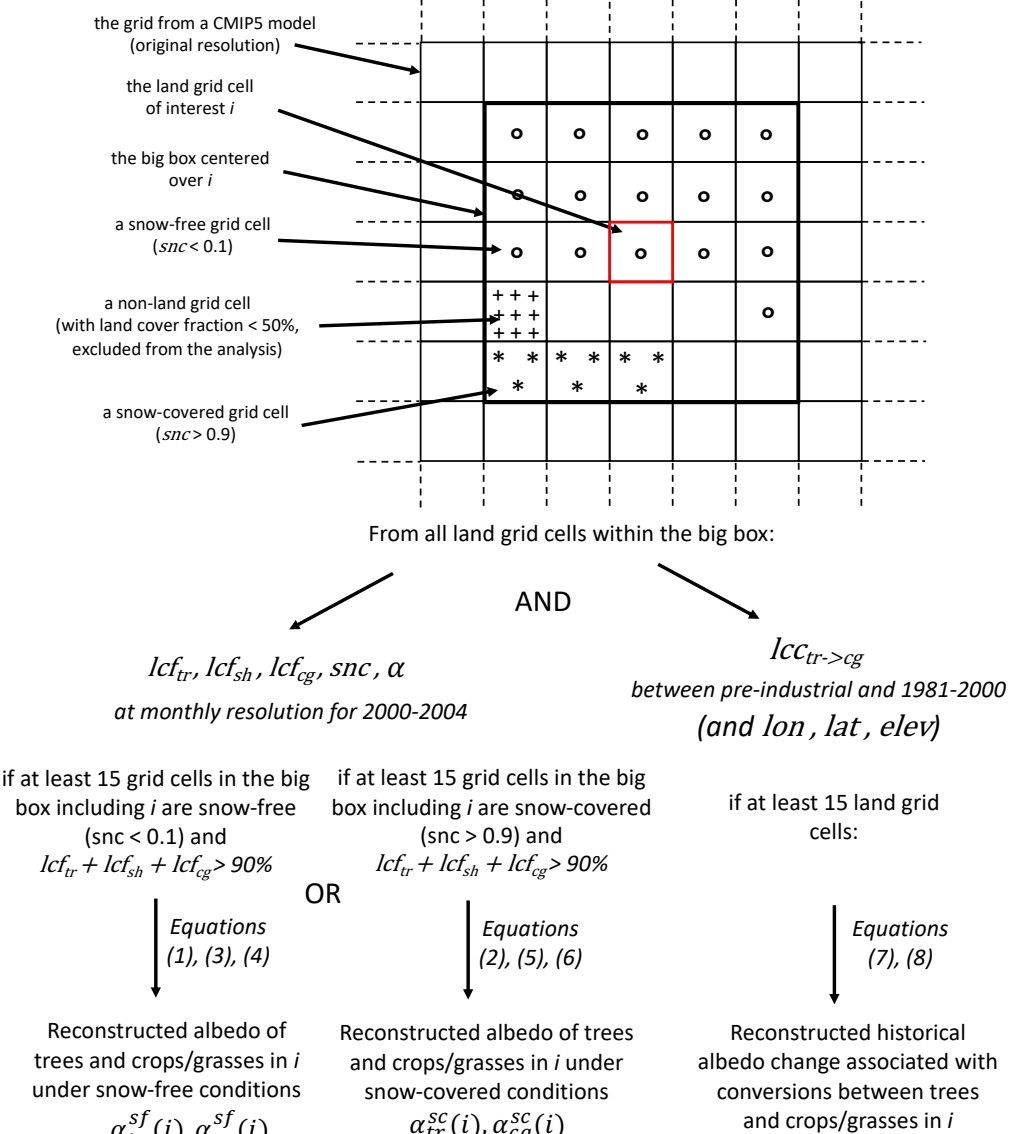

**Figure 1: Description of the two employed reconstruction methodologies.** *snc* stands for snow cover fraction, $\alpha$ for albedo, *lcf* for land cover fraction, *lcc* for land cover conversion, the suffixes *tr*, *sh* and *cg* for trees, shrubs and crops/grasses, respectively, *lon* for longitude, *lat* for latitude, and *elev* for elevation.

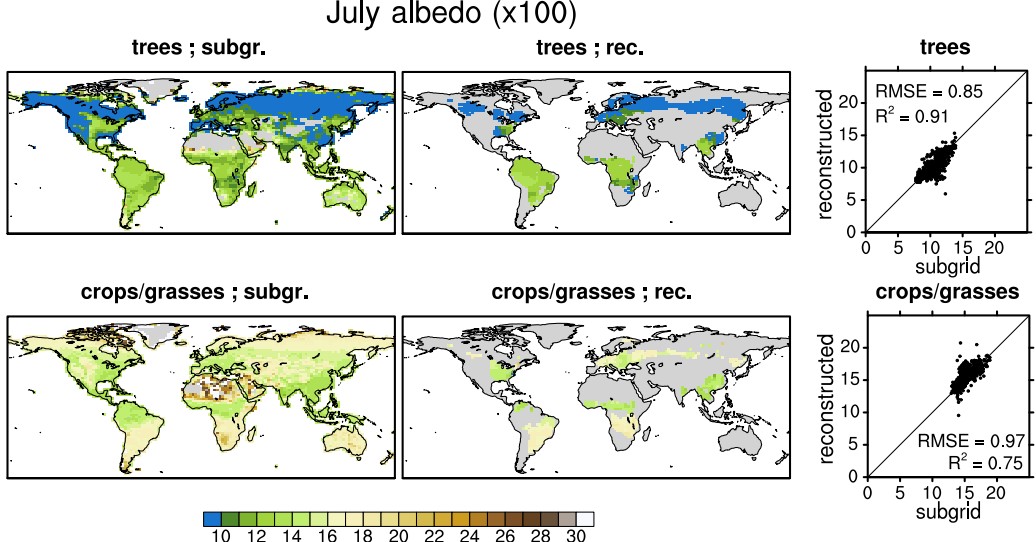

**Figure 2: Subgrid (left) and reconstructed (middle) estimates of the present-day (2002-2010) albedo of trees (upper row) and crops/grasses (lower row) in the CLM4.5 simulations, for the month of July. The scatter plots (right) indicate the relationship between reconstructed (y-axis) and subgrid estimates (x-axis), with each dot indicating the results of a grid cell for which both methods could be applied. Note that albedo values have been multiplied by 100 to facilitate readability.**

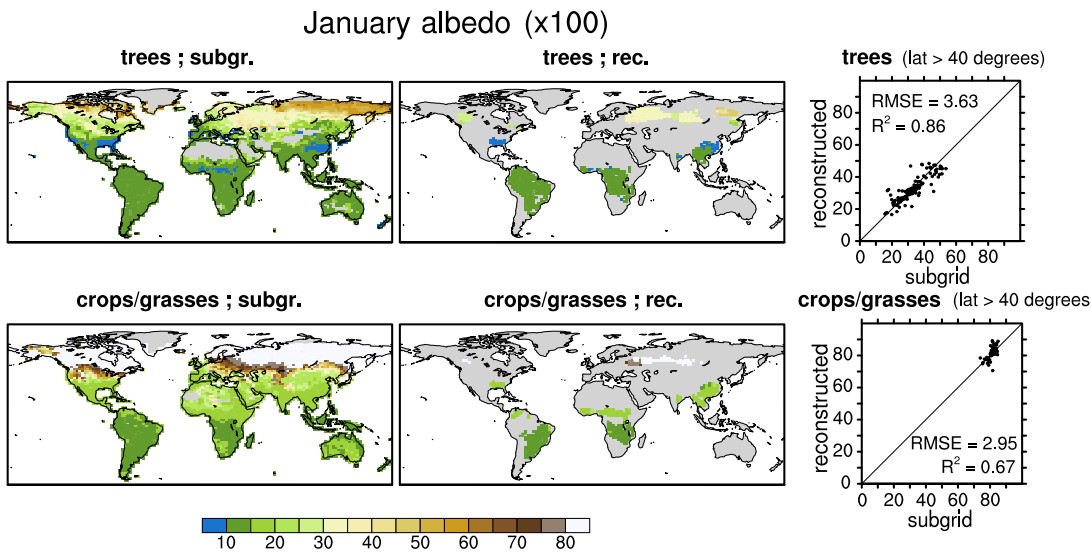

**Figure 3: Same as Figure 2, but for the month of January. The scatter plots in this case only display the results for the grid cells north of 40° (i.e., over areas considered as snow-covered). Note that the scale is different.**

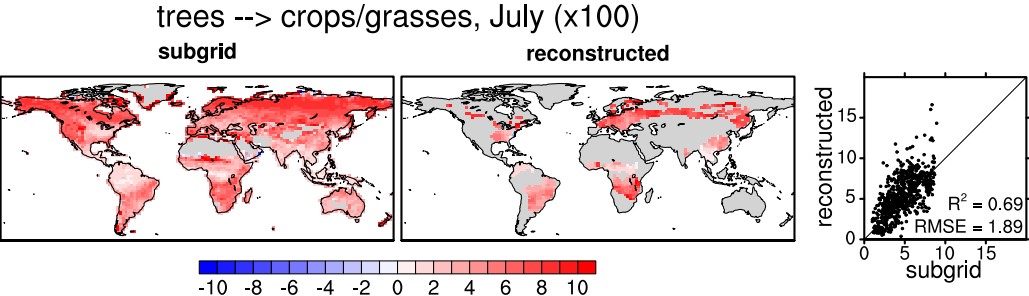

**Figure 4: Subgrid (left) and reconstructed (right) estimates of the present-day (2002-2010) potential albedo change associated to a transition from trees to crops/grasses in the CLM4.5 simulations, for the month of July. Note that absolute differences have been multiplied by 100 to facilitate reading.**

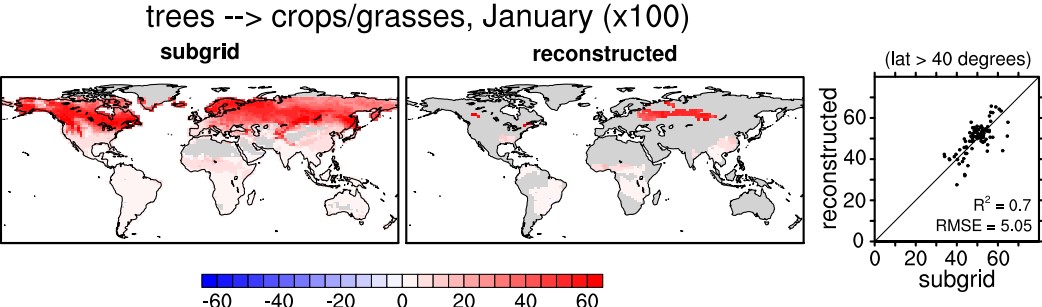

**Figure 5: Same as Figure 4, but for the month of January. The scatter plots in this case only display the results for the grid cells north of 40° (i.e., over areas considered as snow-covered). Note that the scale is different.**

## July albedo trees (X100)

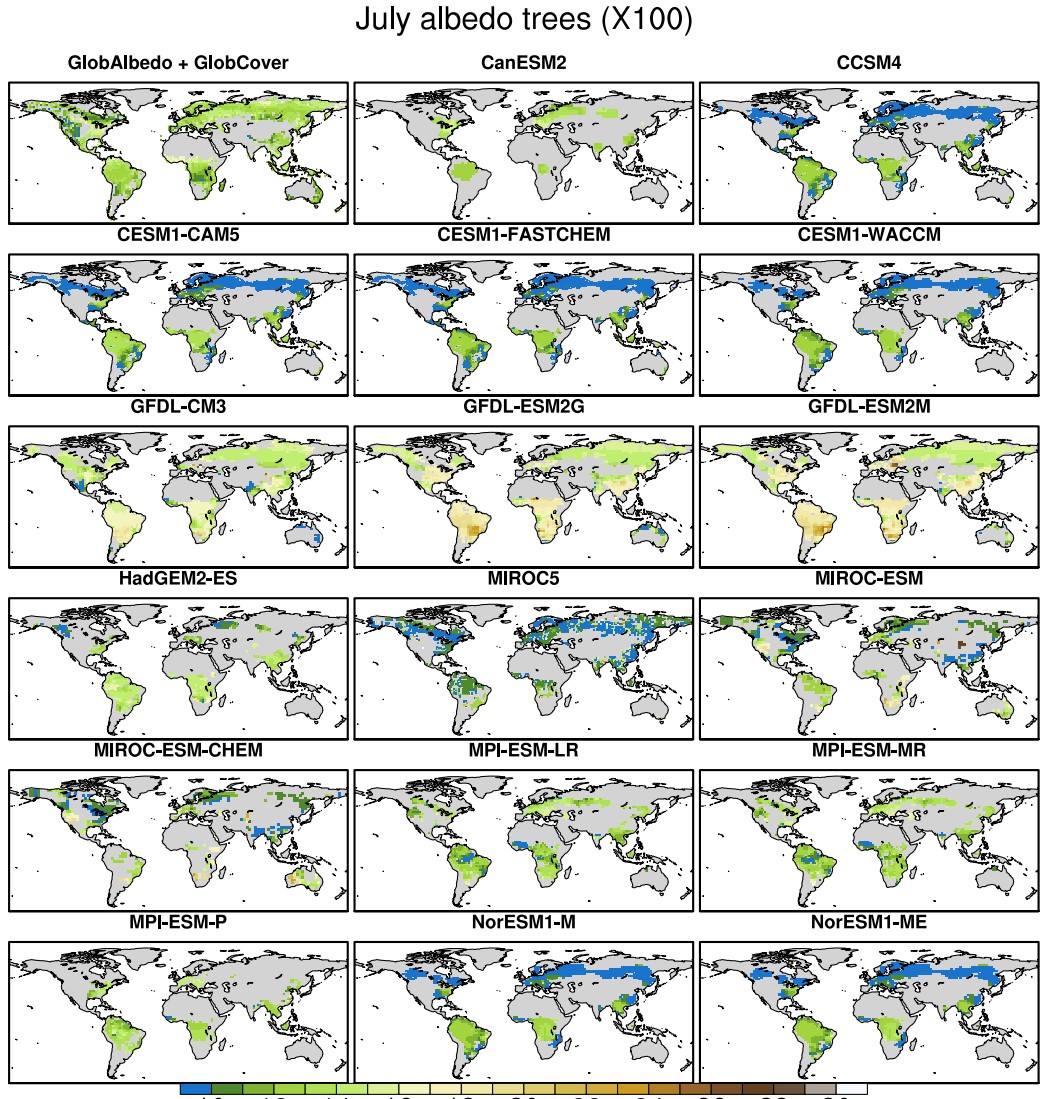

**Figure 6: Present-day July albedo of trees retrieved from the combination of the observational data GlobAlbedo (over the 1998-2011 period) and GlobCover (2005-2006, top-left corner), and in the analysed CMIP5 models (the climatology over the 2000-2004 period was considered). Note that the albedo values have been multiplied by 100 to facilitate readability.**

# January albedo trees (X100)

**Figure 7: Same as Figure 6, but for the month of January. Note that the scale is different.**

## July albedo crops and grasses (X100)

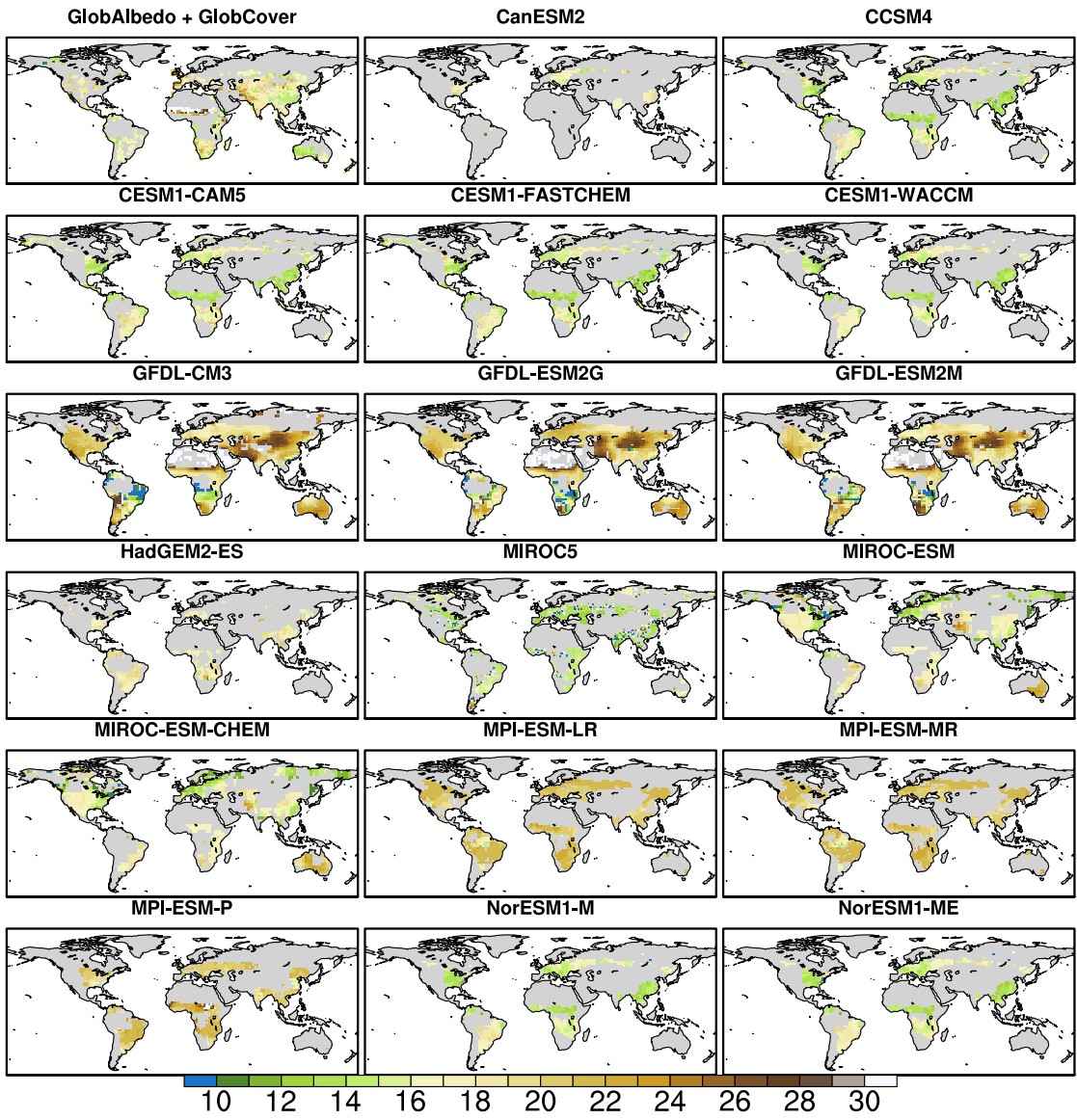

**Figure 8: Present-day July albedo of crops/grasses according to the combined observational data GlobAlbedo (over the 1998-2011 period) and GlobCover (2005-2006, top-left corner), and in the analysed CMIP5 models (the climatology over the 2000-2004 period was reconstructed). Note that the albedo values have been multiplied by 100 to facilitate readability.**

# January albedo crops and grasses (X100)

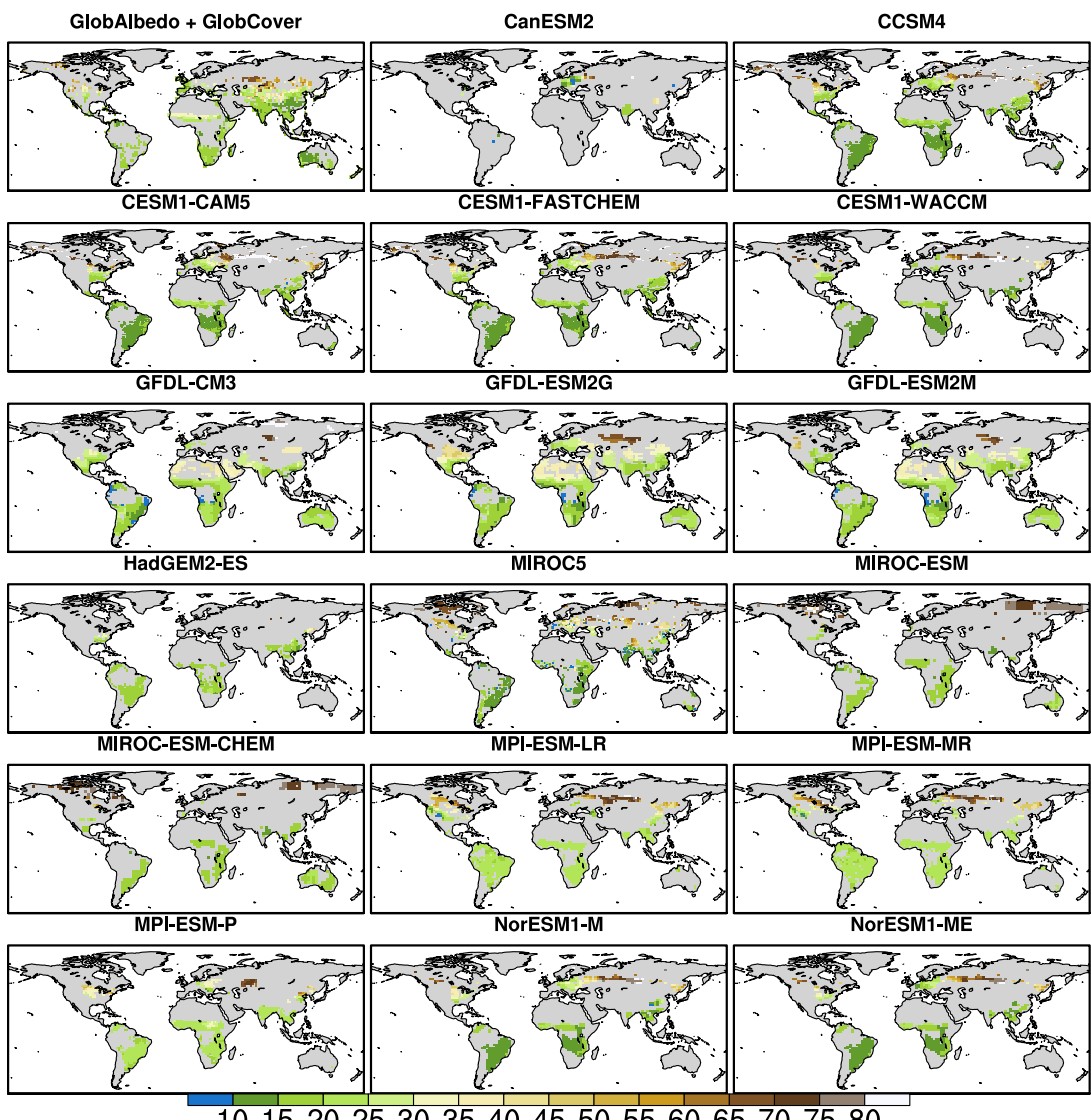

**Figure 9: Same as Figure 8, but for the month of January. Note that the scale is different.**

# July Albedo change trees to crops and grasses (X100)

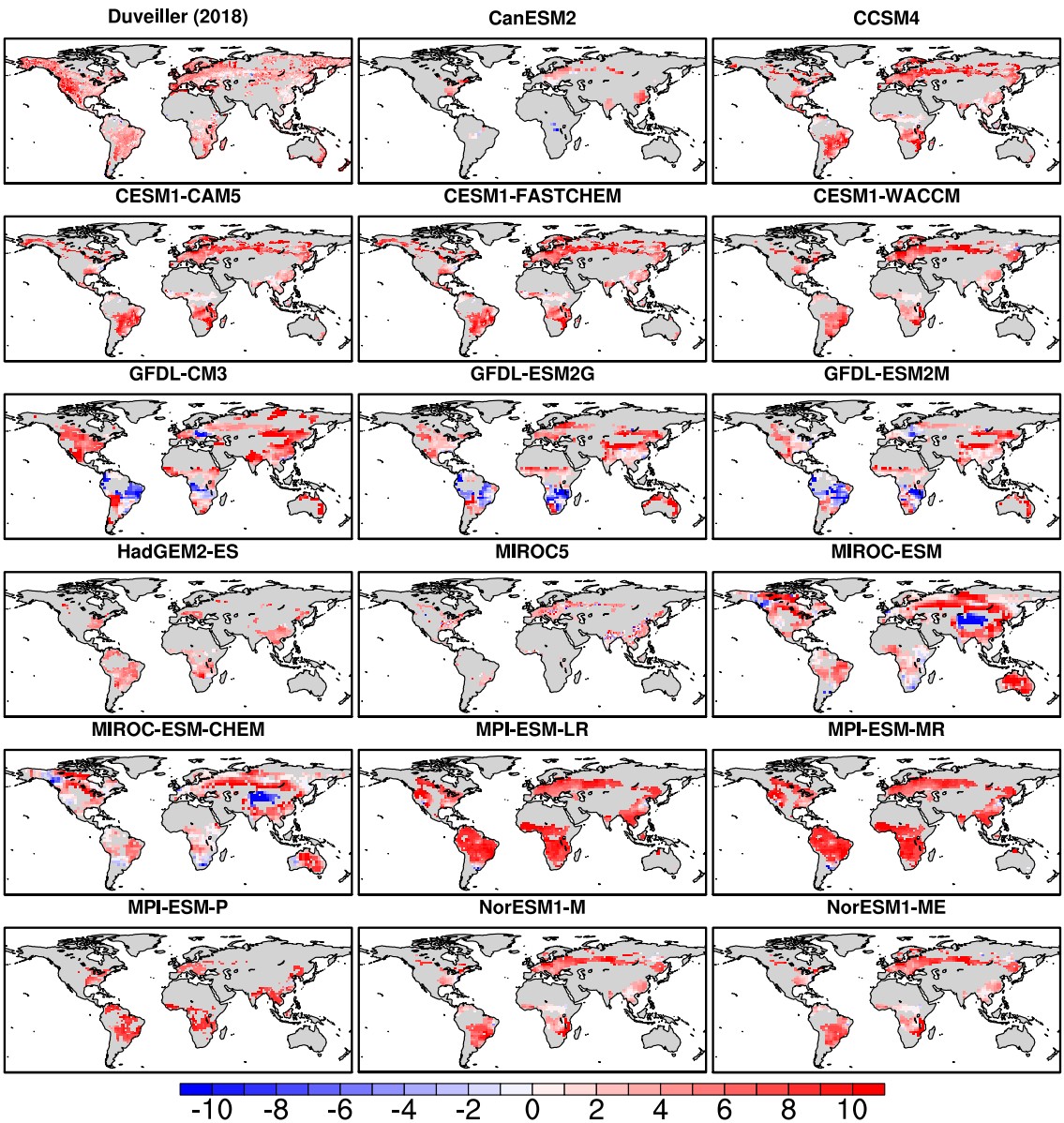

**Figure 10: Potential present-day July albedo change associated to a transition from trees to crops/grasses according to the observational dataset of** Duveiller, Hooker and Cescatti (2018a**, derived from satellite data collected over the 2008-2012 period, top-left corner) and in the analysed CMIP5 models (the climatology over the 2000-2004 period was reconstructed). Note that the absolute differences have been multiplied by 100 to facilitate readability.**

# January Albedo change trees to crops and grasses (X100)

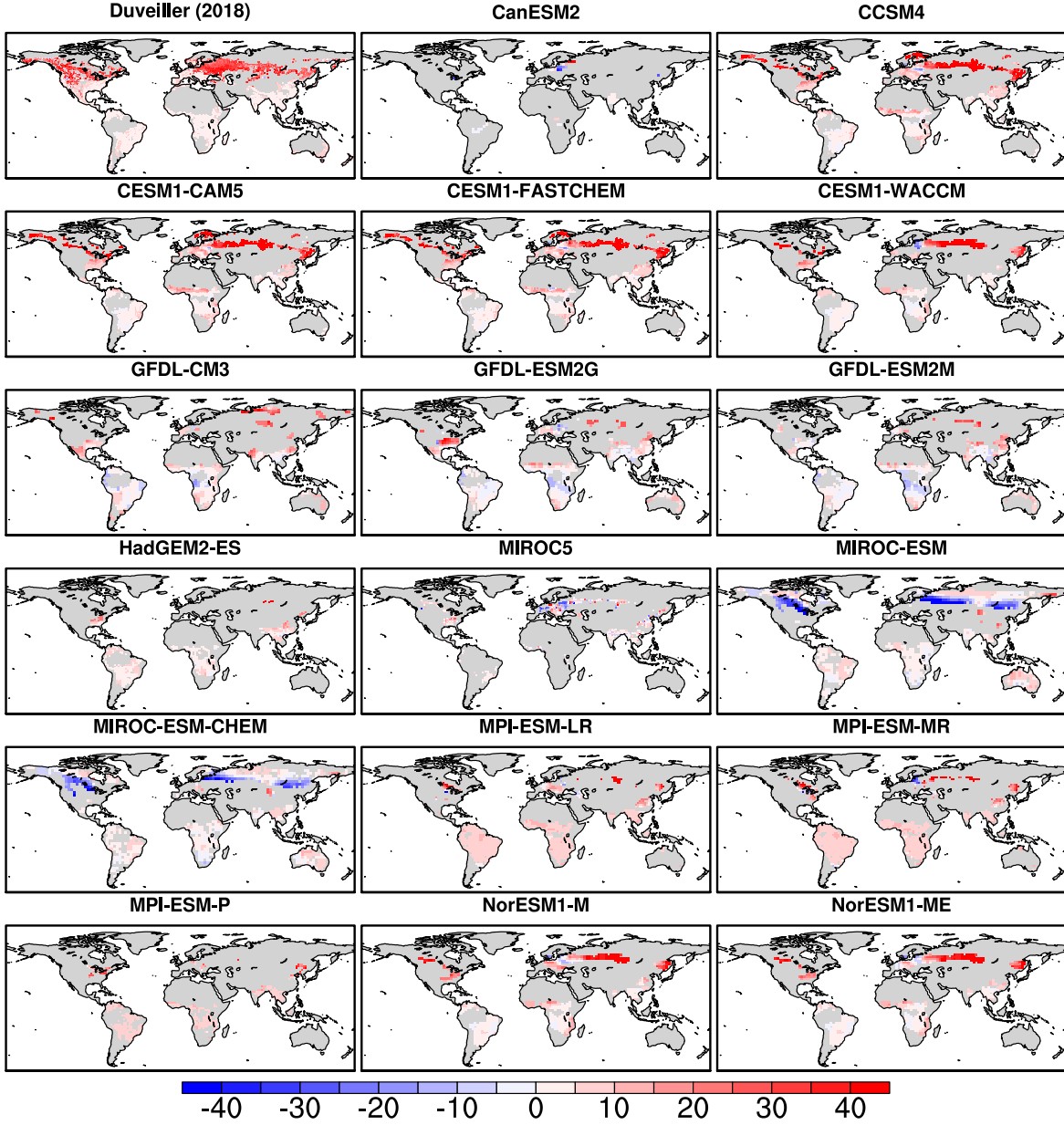

**Figure 11: Same as Figure 10, but for January. Note that the scale is different.**

# Unconstrained RF

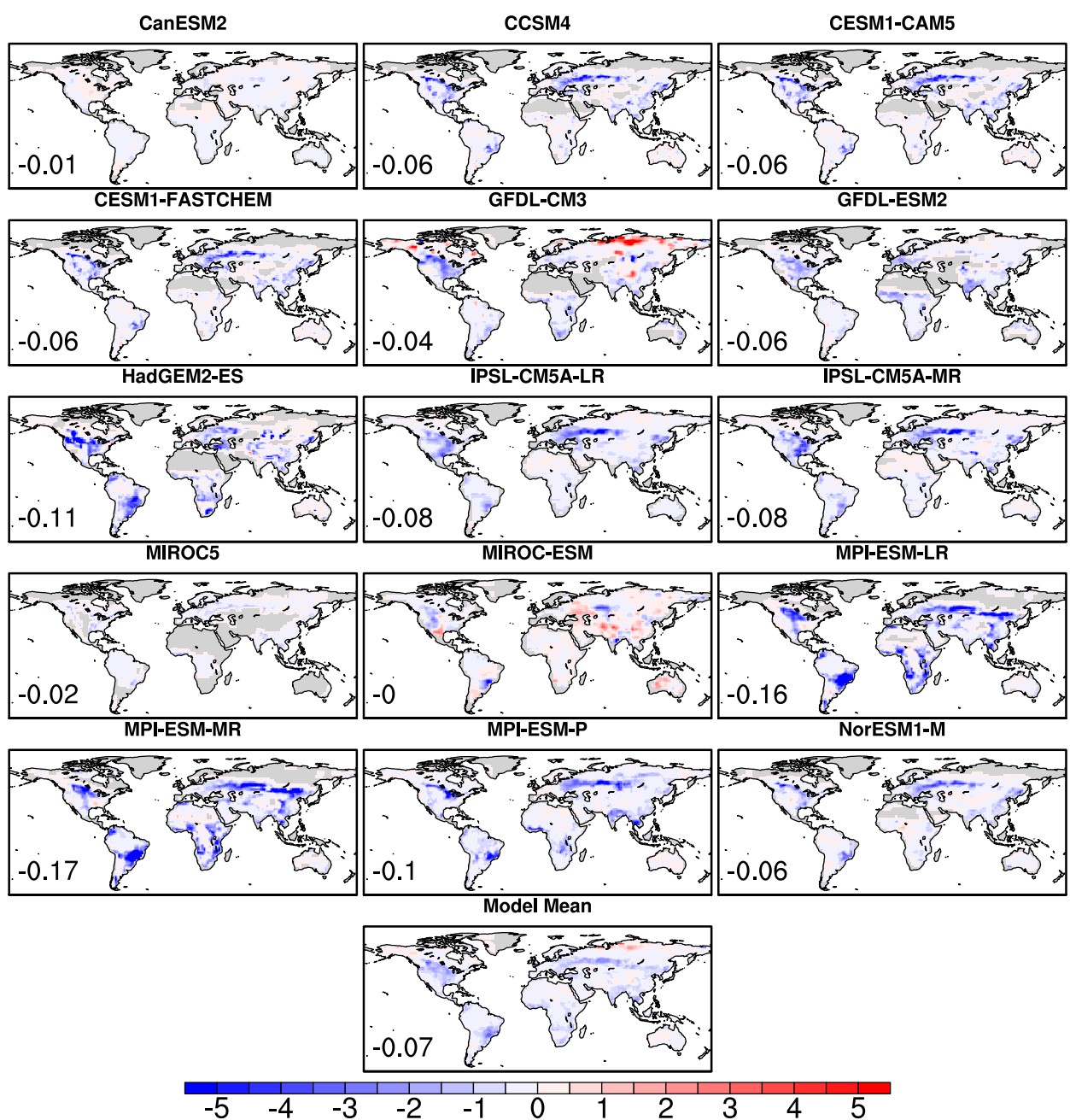

**Figure 12:** **Radiative Forcing from historical conversions between trees and crops/grasses (from the pre-industrial to the 1981-2000 period) in the analysed CMIP5 models (in W/m²), obtained by applying the reconstruction method (see the description of the methodology in Section 2.4). The numbers in the bottom-left corner of each map indicate the global mean Radiative Forcing. For the computation of the Model Mean, if several CMIP5 models contain the same Land Surface Model they were attributed a lower weight so that the sum of their weights equal 1.**

## Constrained RF

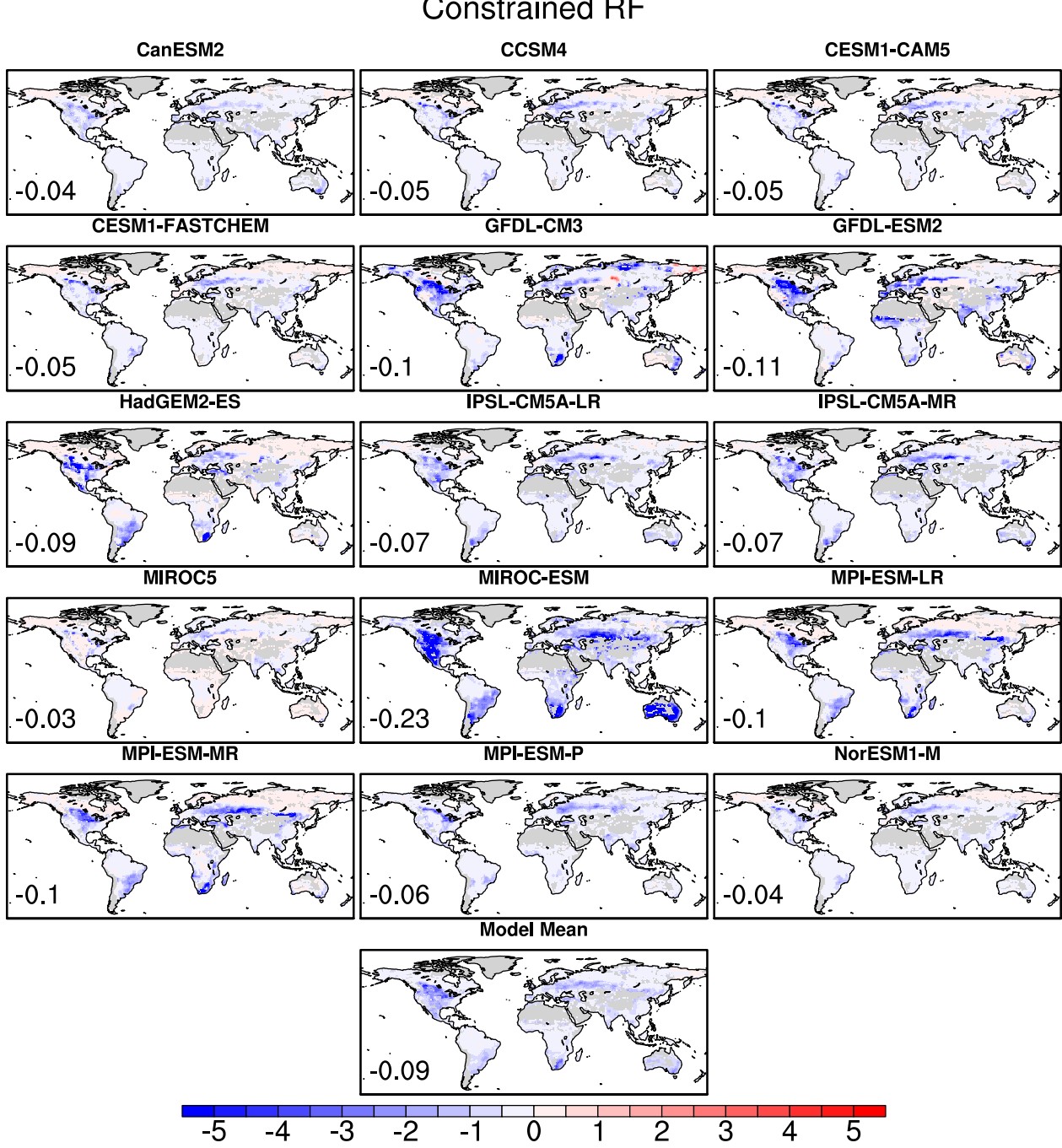

**Figure 13: Same as Figure 12, but for the observation-constrained Radiative Forcing from historical conversions between trees and crops/grasses.**

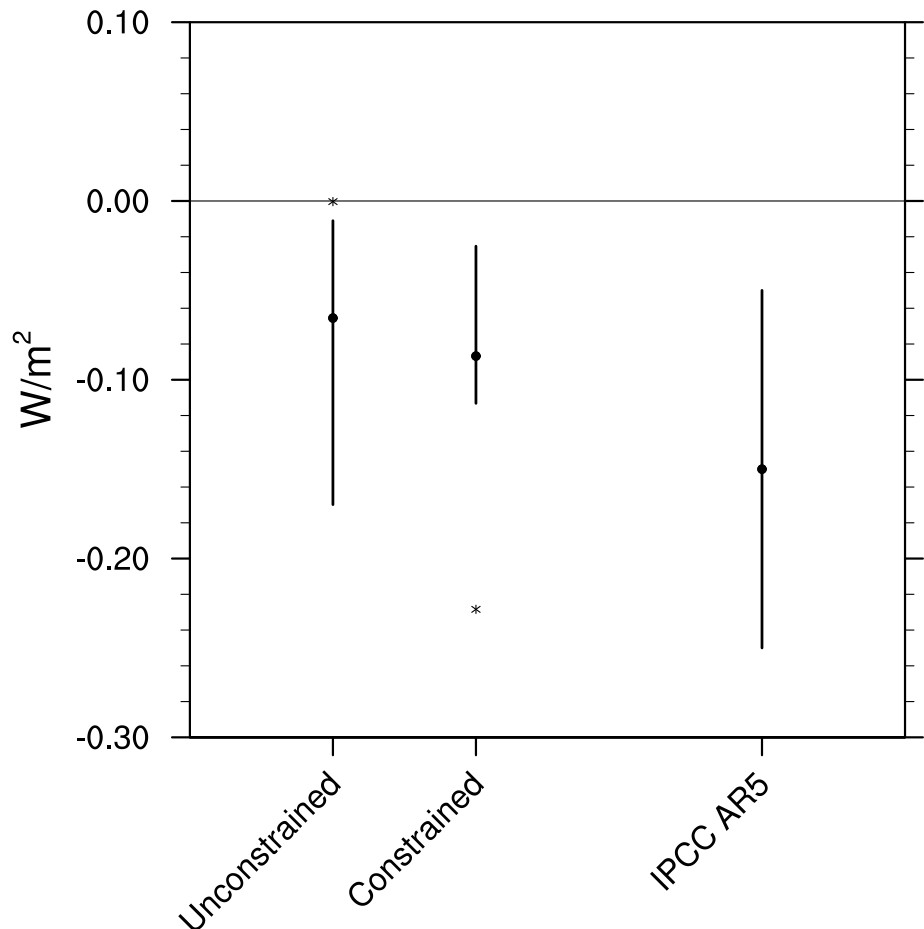

**Figure 14: Spread in the unconstrained (left bar) and observation-constrained (middle bar) estimates of the global Radiative Forcing from historical conversions between trees and crops/grasses (from the pre-industrial to the 1981-2000 periods), for the CMIP5 models shown in Figures 11 and 12 (in W/m²), as well as the IPCC AR5 estimate of the global Radiative Forcing from historical land-use changes (mean estimate and spread as in** Myhre *et al.*, 2013**). The dots on the left and middle bars show the Model Mean**
**results for the unconstrained and observation-constrained estimates, respectively, the asterisks mark the results for the MIROC-ESM model, while the lengths of the bars indicate the spread between the remaining values (i.e., excluding MIROC-ESM).**