# Peer review of "Biases in the albedo sensitivity to deforestation in CMIP5 models and their impacts on the associated historical radiative forcing"

_Earth System Dynamics, 2019_

## Referee Comment (RC1) · Anonymous Referee #1 · 18 Apr 2020

Overall opinion: Lejeune et al. have devised an interesting and innovative method for extracting the albedo of forested and crop/grass land cover types from model simulations and the combination with the space-for-time approach to estimating the effect of land cover change is quite promising. I feel the science is of good quality and the results are useful.

Some questions: I agree with the already posted comment that the RF estimates are based on a parameterization that may contribute its own biases and that the strength of the paper is in the novel albedo methods. I leave it up to the authors whether to address this in the discussion or to change the RF parameterization. Differences in soil type

or texture can affect the albedo of vegetated surfaces and that this would add noise, if not bias to the 'space for time' method. Some brief discussion of the quality, accuracy or uncertainty in the datasets employed (e.g. ESA-CCI) would be helpful. I am curious as to why CRUNCEP V4 was chosen for offline simulations when newer versions and reportedly improved products such as CRUJRA and GSWP3 are available.

Specific minor details:

Line 23: Doesn't constraining something usually reduce its range? Line 26-28: Awkward sentence Multiple locations: "associated to" should be "associated with" Multiple locations: "inferior to" should be "less than" and "superior to" should be "greater than" Line 300: Change "the local albedo difference between albedo and crops/grasses" to "the local albedo difference between forest and crops/grasses".

---

## Short Comment (SC1) · 18 Apr 2020

This is an interesting and timely study. Although its novelty aspects pertain to the surface albedo extraction methods, the authors convert albedo changes to instantaneous radiative forcings (RF) which are subsequently benchmarked to results of several climate modeling studies and to IPCC AR5 estimates. Since the RF quantification and associated benchmarking is made an integral part of the paper, I would encourage the authors to reflect on the uncertainty of their RF estimates, which are based on a very simple parameterization [i.e., Eq. (12) and Cherubini et al. (2012)] that does not account for the spatio-temporal variation in atmospheric optical properties affecting

transmittance of reflected solar radiation.

Bright & O'Halloran (2019), for instance, benchmarked the performance of Eq. (12) to four GCM-based kernels and found persistent positive biases (see f. ex. Figures 1 & 2) and a relative RMSE of about 20% globally. Bright & O'Halloran (2019) proposed a new simplified RF parameterization (see Eq. (17)) that substantially reduces RF "error" (rRMSE of about 6% globally) and made a gridded RF kernel product based on this parameterization freely available (archived here: https://doi.org/10.6073/pasta/d77b84b11be99ed4d5376d77fe0043d8). This product is based on the same underlying CERES v4 data that has been employed in this study and includes uncertainty layers.

I would therefore encourage the authors to either: a) Drop the RF quantification and benchmarking part altogether and keep the focus on the novel albedo methods and merits, or b) Provide a strong justification for using Eq. (12) in light of its uncertainty, or c) Adopt an alternate RF kernel/model that has lower uncertainty.

Bright & O'Halloran 2019:

https://www.geosci-model-dev.net/12/3975/2019/gmd-12-3975-2019.html

---

## Referee Comment (RC2) · Anonymous Referee #2 · 12 May 2020

Summary: The authors attempt to train statistical models to extract the albedo of specific land cover classes in CMIP5 models, with the intent to then calculate the albedo change, and associated radiative forcing (RF), due to deforestation over the historical period. The paper is concise, and reasonably well written, although I found the description of the reconstruction methods to be somewhat unclear. The goals of the research are novel and highly relevant for the land surface and climate modelling communities, and I believe that this work will be suitable for publication once several important concerns are addressed.

Major comments:

[Figure]

-L158: I found the description of the reconstruction method hard to follow, perhaps because several different types of regression models were being applied simultaneously, and perhaps also because non-technical terms like "big box" were introduced. I think a simple diagram, showing the big box and the target (central) cell, and some of the most important quantities involved in the regression models, would be helpful to better explain the methods.

-L235: The authors use an empirical parametrization to relate changes in surface albedo from deforestation to RF at Top of Atmosphere. While this approach is simple and straightforward, I wonder why the authors could not apply a surface albedo radiative kernel instead, as it removes the assumption of temporal and spatial homogeneity in atmospheric transmittance. Given that both surface and TOA clear sky kernels are publicly available, this minor methodological revision would be efficient and most useful. At the very least, the authors could validate their empirical parametrization against a radiative kernel with sample data. The authors are clear to cite the use of such a parametrization in other work. However, given that the the paper attempts to provide a precise, constrained RF estimate from historical deforestation, removing any limitation associated with such a result would provide a significant improvement to the manuscript.

-L284: In Figure 2, the authors show the reconstructed albedo of crops/grasses over northern Eurasia is essentially the albedo of snow (>0.8). Therefore, I take issue with the authors describing their reconstruction as the "extraction of the correct albedo values of specific land cover". This statement is true for the July reconstruction, but the underlying albedo of the vegetation in January is not 0.8, it is most likely very similar to the July value (∼0.2). Therefore, to avoid any potential for confusing the reader, I would like the authors to describe clearly what is being extracted, which is the surface albedo of grid cells with different *underlying* land cover classes".

I note that the same issue also appears to be present in the reconstructed estimate of January albedos in Figures 6 and 8. Therefore, is it possible using this method to

be certain that the albedo change due to LCC (e.g. Figure 10), and associated RF (Figure 11), is properly separated from the albedo change due to changes in snow accumulation and melt over the historical period?

-L311: In Section 3 the authors perform a validation of the reconstruction, and find errors in the reconstructed albedo in the range 10-40%. They conclude at the end of the Section that this method is appropriate to apply to CMIP5 models. But I would have appreciated a little more rigour in this part of the analysis; for example, the authors should define a priori what an acceptable tolerance of error would be. In other words, define what constitutes a "useful" estimate of albedo, which would provide the reader with a stronger basis for interpreting whether 10-40% error is acceptable. Since the focus later is on RF, perhaps one way to define "useful" is in terms of the perturbation that the uncertainty in albedo estimate passes on to the RF calculation, in energy units?

-L321: The authors find considerable intermodel differences in albedo biases, but I couldn't see any discussion linking the different biases to the underlying satellite-derived vegetation datasets used by each modelling group to calibrate their land models. In the case of CanESM2, it's the GLC2000, whereas for models using CLM it is MODIS. Could the authors investigate whether this difference is a contributing factor to the biases? And if so, perhaps the authors could make recommendations to the community as to which datasets produce the lowest biases?

-L335: The discussion of biases in forested albedo when snow is present reminded me of the work by Thackeray et al. (2015, doi=10.1002/2015JD023325), and Wang et al. (2016, doi=10.1002/2015JD023824). I think that citations and connections to these previous studies would be helpful here to explain your results. In addition, Thackeray (2014, doi=10.1002/2014JD021858) shows that the parameterization of canopy albedo in CLM4 was overly sensitive to temperature, resulting in a seasonal cycle that differed significantly from observations.

Minor comments:

-L82 and throughout: I suggest replacing all occurrences of "associated to" –> "associated with".

-L300: "albedo difference between albedo and crops/grasses" –> should this say "between *trees* and crops/grasses"?

-L305: "remain similar" to what?

-L309: I suggest here citing some previous work on computing the "snow-masking effect of forests", for example by Essery (2013).

---

## Referee Comment (RC3) · Anonymous Referee #3 · 5 Jun 2020

This paper presents an approach to diagnosing CMIP5 model outputs with regard to the albedo changes and hence resultant radiative forcing from land cover change from trees to crops/grasses. It borrows the ideas from the analysis of observational data, that is, space for time to reconstruct albedo values of trees and crops/grasses and their differences via an unmixing technique based on linear regression over grid cells within local spatial windows. It compares results among CMIP5 models and between these models and observational data. The evaluation of the reconstruction approach using a model CLM also helps us understand the effectiveness of this approach that has been used to analyze observational data. The study uses both observational and modeling data that varies in terms of native spatial resolutions, temporal resolutions,

and temporal coverages. This diversity in data strengths the investigation but also requires more efforts to achieve a clear and lucid description of methodology and results. I found the description of the methodology and the presentation of the results need some particular improvements. I admit that my knowledge is more on the observational side of studies on land cover and biophysical effects. Some of my questions may have common answers within the hard-core modelling community. Nonetheless, I believe addressing these issues would help the comprehension of the study by a wider audience.

First, Section 2.3.1 needs some clarification in text. I have several questions concerning the understanding of the described method. See detailed comments below. In particular, the line 191-192 states a post-reconstruction estimates of albedo changes by calculating the differences in albedo between trees and crops/grasses. Then the section 2.3.2 looks like a direct estimate of albedo changes from deforestation rate. So, what is the distinction between post-reconstruction estimates and the direct estimates in Section 2.3.2? And which estimates of albedo changes do you present in the results, e.g. Fig. 3, 4, and 9 to 13, and Table 1?

Second, as you use observational and modeling data of different temporal spans/coverages, please specify the year/temporal periods or the temporal coverages for all the figures and tables. For example, Fig 1 & 2, which year/temporal periods are you presenting? In particular for albedo changes, between which year/temporal periods are the presented difference, e.g., Fig. 3 & 4. Almost all the figure/table captions need such clarification. Also, better clarify the spatial resolution of your results. I haven't found explicit statement on the spatial resolution of your reconstructed albedo per land cover class or reconstructed albedo changes per land cover change from trees to crops/grasses. Line 158 says "big boxes of a size of 5 times 5 grid cells", and the grid cells of CMIP5 model outputs you are using have a size of 2 deg (line 90). Does this mean your reconstructed albedo values and albedo changes have a resolution of 10 deg?!

Third, about the radiative forcing from albedo changes. Here you are estimating and showing spatially-explicit RF. I'm not convinced that a single value for k (line 242-243) is enough to account for different solar angles at different latitudes. In Lenton and Vaughan (2009), they were looking into the global effect of geoengineering and a single value of k based on annual global mean transmittance was justified. But I'm convinced it is justified here. Also do you calculate RF from albedo changes month by month? Your presented results seems annual average of RF (fig. 11, 12). Then how do you average monthly changes since your reconstructed albedo changes clearly show monthly differences (fig. 3, 4, 9, 10)

Detailed comments,

1. Line 85-90, Please, even if just in supplementary materials, provide an explicit translation from GlobCover's land cover legend into your trees and crops/grasses for traceability.

2. Line 89, black sky albedo at what solar angle?

3. Section 2.1.2, specify the spatial resolution of D18 data in degrees for easier reading and easier comparison between presented datasets.

4. Line 131, Each grid cell in D18 dataset refers to a specific land cover change, i.e. a pair of land cover classes. What do you mean here by grouping land cover fractions in CMIP5 outputs within one land cover class? Furthermore, D18 provides albedo changes for 45 land cover transitions. How come the consistency with D18 is the reason for focusing just on transition between trees and crops/grasses?

5. Line 165, "inferior to" and "superior to".... maybe just simply say "less than".... ? simple words like "larger than" is enough and better for reading? There are more such cases in the rest of the text to be fixed.

[Figure]

6. Line 184 – 185, What does it mean by "land cover classes are represented" in a grid cell? lcf is larger than zero? If fewer than two grid cells have lcf > 0 for a class, then this class will not be considered in the regression at all. The Eq (2) will have one fewer term on the right side? But will that one grid cell with lcf > 0 (if there is one) for this class be used in such a regression that does not include this class? If so, isn't this inappropriate? If not, please clarify here.

7. Line 186 – 187, 15 is more than half of 25 grid cells in a big box. So in each big box, you can only estimate albedo for either snow-covered land cover classes or snow-free land cover classes. But NOT for both snow-covered and snow-free?

8. Line 187, "where the sum of all the included predictors exceeds 90%.", Please clarify this sentence. Do you mean that at least 90% areal fraction of a big box, that is 0.9*25cells=22.5 cells-equivalent area, is needed for the sum of all the lcf over the included snow-covered OR snow-free grid cells? Or 90% is per EACH grid cell? This sentence reads like the former/first explanation.

9. Section 2.3.2, Eq. (9) and (10), Confusing symbols and texts here. What is difference between the meanings of the $\delta\alpha_{tr\rightarrow cg}$ in Eq. (9) and the $\delta\alpha_{tr\rightarrow cg}(i)$ in Eq. (10)? The $\delta\alpha_{tr\rightarrow cg}$ in Eq. (9) is albedo change as a results of transition from trees to crops/grasses. To me, this means the same as albedo changes due to deforestation that is defined by the $lcc_{tr\rightarrow cg}$, transition rate from trees to crops/grasses. If $\delta\alpha_{tr\rightarrow cg}$ in Eq (9) is a known quantity since you need it for the regression and it means the albedo change due to deforestation, what is the physical rational of the Eq (9)? And what is the point of using regression to achieve the estimate by Eq. (10)? And how did you get the value of $\delta\alpha_{tr\rightarrow cg}$ on the left side of the Eq. (9)? Is it from the methods given in Section 2.3.1 line 191-192? Please clarify.

10. Line 218, "a jackknife resampling is conducted: Alternatively, and as", some typo here? esp. about the weird punctuation marks?

11. Line 248, the $\delta\alpha_{tr\to cg}$ is estimated per month. Do you also estimate RF per month?

12. Line 250-251, it reads very unclear even considering the preceding texts and the context. Please elaborate on this.

13. Section 2.5, no information about how you estimated albedo changes of land cover change from trees to crops/grasses in this subgrid experiment. But your results presented such subgrid estimates of albedo changes (figure 3&4, table 1).

14. Line 263, what do you mean by "pixel" here? A grid cell in the CLM simulation? If so, be consistent in the terminology.

15. Line 263 – 265, Do you differentiate snow-covered and snow-free fractions of a grid cell when you calculate albedo values for trees or crops/grasses in this area-weighted average? If so, can you provide some explanation here?

16. Section 3.2, Which estimate of albedo changes are you presenting here, the estimate by Section 2.3.1, line 191-192, or the estimate by Section 2.3.2?

17. Line 464, what information in "for which this information is available"? This last part reads redundant and only adds confusion to this sentence.

18. Fig. 1 caption, what's the absolute difference here? You mean albedo values?

19. Almost all the figures of maps, good to have the maps here for spatial comparison. But it is only qualitative. Can you present a scatter plot over common grid cells between subgrid estimates and reconstructed estimates?

20. Fig. 10 v.s. Fig. 11, why the different sets of models for albedo changes and RF?

21. Fig. 11, why not present a model mean as fig. 12?

---

## Author Comment (AC1) · 13 Jul 2020

Thank you for the comment. We have decided to follow the suggestion to use the CERES-based albedo change albedo kernel (CACK) from Bright and O'Halloran (2019) for the Radiative Forcing calculations. As a result, the values of global RF associated with historical conversions between trees and crops/grasses are systematically less negative by ∼20-30% for each of the CMIP5 models considered in Section 5 (see the revised Figures 11 and 12 in attachment).

[Figure]

**Fig. 1.**

[Figure]

**Fig. 2.**

---

## Author Comment (AC2) · 16 Jul 2020

Thanks to the reviewer for the useful and overall positive comments.

As specified in the response to Ryan Bright's comment, we have decided to follow his suggestion to use the CERES-based albedo change albedo kernel (CACK) from Bright and O'Halloran (2019) for the Radiative Forcing calculations. As a result, the values of global RF associated with historical conversions between trees and crops/grasses are systematically less negative by ∼20-30% for each of the CMIP5 models considered in Section 5 (see the revised Figures 11 and 12 in attachment).

[Figure]

Surface albedo is indeed influenced by both the vegetation canopy and the soil, and this is the case in both satellite-derived observational products and climate models. It is true that both the vegetation canopy and the soil can exhibit variations in solar reflectance even for a same land cover, e.g. due to variations in Leaf Area Index or in soil texture. If such variations occur within a 'big box' of 5X5 grid cells, this can indeed introduce noise to the reconstruction methodology, and thus explain a substantial part of the RMSE of the reconstruction methodology discussed in Section 3.

When preparing the revised manuscript, we have added brief discussions of the quality of the employed observational datasets. The GlobAlbedo and MODIS MCD43C3 albedo products are considered to be of very good quality overall and show good agreement (global R2 of 0.85). Some problems associated with snow detection were identified in GlobAlbedo but the resulting artifacts are most significant at very high latitudes (>70°), which are of lesser interest for our study (Muller et al, 2013). Imprecisions in land cover datasets such as GlobCover and ESA-CCI may especially arise via misclassification between land cover types within the broad trees or crops/grasses classes (e.g., between two types of trees) or the difficulty to properly identify medium-sized or mixed-type vegetation (i.e., shrub or savanna-like). In contrast, because these products are best at distinguishing very distinct land cover types such as trees and crops/grasses, the satellite-derived albedo values of these two broad classes (retrieved following the methodology presented in Section 2.1.1) as well as their differences (obtained from the D18 data) are characterised by relatively low uncertainties.

At the time the CLM4.5 simulation was conducted, CRUNCEP V4 was the recommended forcing data set (Chapter 26 in Oleson et al., 2013). Indeed, CLM offline simulations forced by GSWP3 represent surface albedo better than simulations forced by CRUNCEP (http://www.cesm.ucar.edu/models/cesm2/land/). However, the main purpose of this simulation is to demonstrate that the reconstruction method retrieves similar albedo alterations due to land-cover changes as the subgrid method. Therefore, the performance of this particular simulation is no major concern, as long as the simulated

albedo is realistic to a sufficient extent.

Line 23: "Constraining" the global RF estimates by using the albedo changes due to conversions between trees and crops/grasses from satellite data instead of the individual CMIP5 models indeed leads to a somewhat unexpected increase in the model range. This result occurs because of two models which exhibit unrealistic historical conversion rates from trees to crops/grasses, which we therefore decided to discard for the final estimation of the RF from historical land-cover changes. This second step can also be considered as an observational constraint, and we can modify the corresponding sentence in the abstract accordingly.

Lines 26-28 and 300: We will take these remarks concerning the language into account when proofreading the manuscript again.

References: Muller, J.-P. et al. (2013) GlobAlbedo Final Product Validation Report. Oleson, K. W. et al. (2013) 'NCAR/TN-503+STR NCAR Technical Note Technical Description of version 4.5 of the Community Land Model (CLM) Coordinating Lead Authors'. Available at: http://library.ucar.edu/research/publish-technote (Accessed: 31 December 2019).

[Figure]

**Fig. 1.** Revised Figure 11.

[Figure]

**Fig. 2.** Revised Figure 12.

---

## Author Comment (AC3) · 16 Jul 2020

We thank the reviewer for taking the time to go through the manuscript and submitting detailed comments. We are providing answers to these below, by referring to the individual comments through mentions of the same line numbers given by the reviewer. We also attach a point-by-point response including the reviewer's comments, which may facilitate the second round of reviews.

-L158: We have extensively worked on improving the methodology description. Including a figure was indeed a good idea to facilitate its understanding by the reader, and we have followed this piece of advice from the reviewer. The new Figure 1 (see be-

low) should clarify non-common technical terms like "big box", the used reconstruction methods and some methodological steps we apply to increase their reliability.

-L235: All reviewers have suggested to use a different kernel to convert the reconstructed historical albedo changes due to conversions between trees and crops/grasses into RF estimates. Following Ryan Bright's comment, we have decided to use the version 1.0 of the CERES-based albedo change kernel (CACK) from Bright and O'Halloran (2019) for the Radiative Forcing calculations. This kernel is based on a novel, simplified parameterisation of shortwave radiative transfer and driven with downwelling shortwave radiation values at the surface and the top of the atmosphere obtained from the Clouds and the Earth's Radiant Energy System (CERES) Energy Balance and Filled (EBAF) 1°-resolution products. CACK was evaluated by Bright and O'Halloran (2019): While being more easily understandable and easier to apply than kernels derived from climate models, it is able to mimic them more faithfully than five previously employed analytical, semi-empirical and empirical kernels.

-L284: The reviewer is right that the surface albedo in both observational data and climate models is influenced by both the vegetation canopy and the soil reflectance. We have now clarified this at the beginning of Section 2.1.1. For the sake of simplicity, we however use the formulation "albedo of a specific land cover class" when referring to this mixed contribution of the soil and canopy to the surface albedo. This has also made clear in Section 2.1.1.

We would like to stress that the present-day albedo of trees and crops/grasses is only reconstructed following the method described in Section 2.3.1 in order to be evaluated against satellite-derived data, as discussed in Section 4 and illustrated in Figures 5-10. In contrast, the historical albedo changes associated with transitions between trees and crops/grasses between the pre-industrial and 1981-2000 periods are reconstructed following the method described in Section 2.3.2, so that the associated global RF can be derived and discussed in Section 5 (based on Figures 11 and 12) in light of the model biases identified using the first reconstruction method. We acknowledge that this may

have been ambiguous in the submitted manuscript, and intend to make it clearer in the revised version. That being said, changes in surface albedo over vegetated surfaces between the pre-industrial and present-day periods are mainly influenced by changes in albedo of the vegetation canopy, in the fraction of ground (soil or snow) that is shed from sunlight by the vegetation canopy, and in the albedo of the ground. The first two contributions are mostly influenced by LCC and in particular transitions between trees and crops/grasses. They are therefore included in the term $\delta 1$ of Equation (9). The latter contribution is mostly influenced by other climate forcings such as greenhouse gases, whose influence has a larger spatial extent which is thus assumed to be constant across a big box and included in the term $\delta 0$. For the models for which factorial experiments (with LCC only or with all forcings except LCC) are available, we were able to directly extract the simulated change in surface albedo due to LCC. The similarities between these direct estimates and the results from the reconstructed method (compare the columns on the left and right sides of Figures S6, S7, S11 and S13) confirm the ability to properly separate surface albedo changes due to LCC from those due to changes in snow accumulation and melt. Some differences are found for the GFDL-ESM2 model but both estimates remain compatible given the uncertainty ranges of each method (see Figure S21).

-L311: the present-day albedo of trees and crops/grasses that is reconstructed in the CMIP5 simulations using the methodology described in Section 2.3.1 is not used later on in the RF calculations, but simply to be evaluated against reference satellite-derived data. This evaluation effort reveals that, in some of the analysed CMIP5 models, the reconstructed albedo changes associated with transitions from trees to crops/grasses can differ from the reference values from Duveiller et al. (2018) by $\sim$0.05 (respectively, $\sim$0.4) over snow-free (respectively, snow-covered) areas (see Figures 9 and 10). These differences being substantially higher than the RMSE of the reconstruction (which amounts to $\sim$0.019 over snow-free and $\sim$0.051 over snow-covered areas), we affirm that this RMSE is acceptable enough for the purpose of the conducted model evaluation.
-L321: Because our analysis focuses on the potential albedo change resulting from a land cover conversion between trees and crops/grasses rather than the mean surface albedo of each model grid cell, differences in the vegetation distributions of individual models should play a limited role in the biases identified in Figures 5-10. It is however true that if a model has a too low proportion of trees in a given region, for example, it can hinder the retrieval of the albedo of trees in this same region and therefore limits the scope of our analysis.

-L335: The papers by Thackeray et al. (2014, doi=10.1002/2014JD021858 and 2015, doi=10.1002/2015JD023325), as well as Wang et al. (2016, doi=10.1002/2015JD023824) brought to our attention by the reviewer are indeed very relevant for the interpretation of our results. They also point at model deficiencies which can be linked to some of the biases identified in our study, such as the too high albedo of trees in snow-covered areas in the MIROC5 model.

We will also take the minor comments concerning the language into account when proofreading the manuscript again.

Bright, R. M. and O'halloran, T. L. (2019) 'Developing a monthly radiative kernel for surface albedo change from satellite climatologies of Earth's shortwave radiation budget: CACK v1.0', Geosci. Model Dev, 12, pp. 3975–3990. doi: 10.5194/gmd-12-3975-2019. Duveiller, G., Hooker, J. and Cescatti, A. (2018a) 'A dataset mapping the potential biophysical effects of vegetation cover change', Scientific Data. Nature Publishing Groups, 5. doi: 10.1038/sdata.2018.14.

[Figure]

the grid from a CMIP5 model
(original resolution)

the land grid cell
of interest $i$

the big box centered
over $i$

a snow-free grid cell
($snc < 0.1$)

a non-land grid cell
(with land cover fraction < 50%,
excluded from the analysis)

a snow-covered grid cell
($snc > 0.9$)

From all land grid cells within the big box:

$lcf_{tr}, lcf_{sh}, lcf_{cg}, snc, \alpha$

*at monthly resolution for 2000-2004*

$lcc_{tr->cg}$

*between pre-industrial and 1981-2000*
*(and $lon$, $lat$, $elev$)*

if at least 15 grid cells
are snow-free
(snc < 0.1) and
$lcf_{tr} + lcf_{sh} + lcf_{cg} > 90\%$

OR

if at least 15 grid cells are
snow-covered
(snc < 0.1) and
$lcf_{tr} + lcf_{sh} + lcf_{cg} > 90\%$

if at least 15 land grid
cells:

*Equations
(1), (3), (4)*

*Equations
(2), (5), (6)*

*Equations
(7), (8)*

Reconstructed albedo of
trees and crops/grasses in $i$
under snow-free conditions
$\alpha_{tr}^{sf}(i), \alpha_{cg}^{sf}(i)$

Reconstructed albedo of trees
and crops/grasses in $i$ under
snow-covered conditions
$\alpha_{tr}^{sc}(i), \alpha_{cg}^{sc}(i)$

Reconstructed historical
albedo change associated with
conversions between trees
and crops/grasses in $i$

Figure 1: Description of the two employed reconstruction methodologies.
*snc* stands for snow cover fraction, $\alpha$ for albedo, *lcf* for land cover fraction,
*lcc* for land cover conversion, the suffixes *tr*, *sh* and *cg* for trees, shrubs and
crops/grasses, respectively, *lon* for longitude, *lat* for latitude, and *elev* for
elevation.

**Fig. 1.** Suggestion for a new Figure 1 describing the methodology.

---

## Author Comment (AC4) · 17 Jul 2020

Many thanks to the reviewer for taking the time to go through the manuscript and submitting such detailed comments. This is very helpful to improve the manuscript. We are providing answers to the reviewer's comments below.

- In response to the first main comment: We indeed introduce two different reconstruction methods in the manuscript. We aim at clarifying the methodology section in a revised version of the manuscript, and especially at explaining better what these two methods intend to do and how. We also hope that a newly included figure (see below) will provide visual help in that respect. In this study we reconstruct two different quantities in CMIP5 models: 1) the simulated present-day albedo of trees and crops/grasses, to evaluate the albedo change arising from a potential transition between these two classes against observational data, and 2) the historical surface albedo changes associated with transitions between trees and crops/grasses, followed by an assessment of their consequence in terms of Radiative Forcing. Based on Table 1 and Figures 1-4, Section 3 focuses on the evaluation of the ability of the first employed reconstruction method to extract the first quantity (simulated present-day albedo of trees and crops/grasses) in CMIP5 all-forcings simulations. In Section 4, based on Figures 5-10 we evaluate the albedo change arising from a potential transition between trees and crops/grasses in CMIP5 models against observational data. This quantity has been extracted using the first reconstruction method, which has previously been evaluated in Section 3. In Section 5, based on Figures 11-13 we discuss the historical surface albedo changes associated with transitions between trees and crops/grasses between the pre-industrial and 1981-2000 periods, and which have been reconstructed in CMIP5 models using the second reconstruction method.

- In response to the second main comment: We will pay attention at including the spatial and temporal resolutions as well as the temporal coverages where appropriate in the legends of the figures and tables, as suggested by the Referee. Moreover, the revised methodology section as well as the new figure (see below) should make clearer that within one big box, we reconstruct the albedo values (or albedo changes) for the grid cell in the center of the big box, i.e. the reconstructed albedo values and albedo changes have the original model resolution (about 2°).

- In response to the third main comment: All reviewers have suggested to use a different kernel to convert the reconstructed historical albedo changes due to conversions between trees and crops/grasses into RF estimates. Following Ryan Bright's comment, we have decided to use the version 1.0 of the CERES-based albedo change kernel (CACK) from Bright and O'Halloran (2019) for the Radiative Forcing calculations. This kernel is based on a novel, simplified parameterisation of shortwave radiative trans-

fer and driven with downwelling shortwave radiation values at the surface and the top of the atmosphere obtained from the Clouds and the Earth's Radiant Energy System (CERES) Energy Balance and Filled (EBAF) 1°-resolution products. CACK was evaluated by Bright and O'Halloran (2019): While being more easily understandable and easier to apply than kernels derived from climate models, it is able to mimick them more faithfully than five previously employed analytical, semi-empirical and empirical kernels. CACK provides a monthly climatology, it is thus possible to compute a change in top-of-the-atmosphere net radiation amounts due to surface albedo changes that vary from month to month by summing up the contributions from each month. This is summarized in the equation for which a screenshot is attached to this reply (see below).

Detailed comments,

1. As asked by the reviewer, we have included an explicit translation from Glob-Cover's land cover classification into the two broad classes that we used (trees and crops/grasses) in a new Table S1, and added a reference to this Table in Section 2.1.1.

2. We understand that the GlobAlbedo product makes use of an optimal estimation approach including angular integrals and a gap-filling technique based on the MODIS surface anisotropy dataset in order to integrate data derived from the Advanced Along-Track Scanning Radiometer (AATSR), SPOT4-VEGETATION, SPOT5-VEGETATION2, and MERIS instruments, which exhibit different spectral and angular sampling. We have included this information in Section 2.1.1.

3. The resolution of the dataset from Duveiller et al. (1°) was missing and has now been specified in Section 2.1.2.

4. As specified in Section 2.1.2, we have used the version of the D18 dataset that provides albedo changes for only six land cover transitions between four broad land cover classes (forests, shrubs, crops/grasses and savannas). This classification scheme is referred to as IGBPgen in the paper describing the dataset. Consistently with this

scheme, when represented in the CMIP5 models we have considered grasses, crops and pasture as belonging to one single class: crops/grasses.

5. We will take the comments concerning the language into account before submitting the revised manuscript.

6. The sentence containing "land cover classes are represented in a grid cell" will be reformulated as such: "Therefore, each predictor (*lcftr*, *lcfsh*, *lcfcg*) is only included in the regression (i.e., its corresponding term is included in Equation (1) or (2)) if its value is greater than 0 in at least two snow-free (if *i* is snow-free) or snow-covered grid cells (if *i* is snow-covered)." If for example *lcfsh* is not included in a regression, that one grid cell with *lcfsh* > 0 (if there is one) may still be considered in the regression if over that grid cell *lcftr* + *lcfcg* > 90%. This would effectively mean that lcfsh is less than or equal to 10%, therefore that the albedo of shrubs only accounts for a small portion of the mean surface albedo over that grid cell, which thus justifies the consideration of that grid cell in the regression.

7. The referee is correct that if a grid cell is snow-free (respectively, snow-covered) in a given month, we only estimate albedo for snow-free (respectively, snow-covered) conditions. We have reformulated parts of Section 2.3.1 to clarify this point, and the new figure should also help in that respect.

8. We suggest to reformulate the sentence originally present line 187 as such: "Moreover, the regressions are only conducted in the big boxes with a at least 15 grid cells (either snow-free or snow-covered) in which the sum of all the included predictors exceeds 90%." We hope that the inclusion of a new figure will also help to make this methodological point clearer.

9. There was indeed a mistake, there should have been no subscript on the term on the left side of Eq. (9).

10. The description of the jackknife resampling has been reformulated.

11. We indeed estimate the instantaneous RF resulting from historical conversions from trees to crops/grasses for each month, then compute the annual mean value. We hope that the revised methodology description and in particular the equation for which a screenshot is attached to this reply (see below).

12. In an attempt to clarify the text lines 250-251, we will now differentiate between the potential surface albedo change associated with a transition from trees to crops/grasses ($\delta\alpha_{tr\rightarrow cg}$, reconstructed using the method described in Section 2.3.1) and historical surface albedo changes due to conversions between trees and crops/grasses between the pre-industrial and 1981-2000 periods ($\Delta\alpha_{tr\rightarrow cg}$, reconstructed using the method described in Section 2.3.2). New Equations (11) and (12) will also be included, which should help for this clarification.

13. We have tried to formulate a clearer explanation of how the subgrid estimates of the present-day albedo of trees and crops/grasses are extracted from the CLM4.5 simulations. Especially, the following sentences should help to understand this point: "Surface albedo values were output for each tile in these simulations, enabling to extract a subgrid albedo value for each land cover class (trees or crops/grasses, similarly as in Malyshev et al., 2015; Meier et al., 2018). For each grid cell and each month, the albedo values for these two land cover classes are computed as the area weighted mean albedo across each PFT pertaining to the respective class over the analysis period. This reference value, later referred to as "subgrid" estimate, can then be compared to the reconstructed albedo values." Malyshev, S. et al. (2015) 'Contrasting Local versus Regional Effects of Land-Use-Change-Induced Heterogeneity on Historical Climate: Analysis with the GFDL Earth System Model', Journal of Climate, 28, pp. 5448–5469. doi: 10.1175/JCLI-D-14-00586.1. Meier, R. et al. (2018) 'Evaluating and improving the Community Land Model's sensitivity to land cover', Biogeosciences, 15, pp. 4731–4757. doi: 10.5194/bg-15-4731-2018.

14. "Pixel" will be changed to "grid cell" to ensure consistency across the manuscript and more clarity for the reader.

15. We don't differentiate between snow-covered and snow-free grid cells when looking at the subgrid albedo values for trees and crops/grasses, but extract the subgrid albedo values for any snow cover fraction.

16. In Section 3 we focus on the evaluation of the ability of the first employed re-construction method (presented in Section 2.3.1) to extract the simulated present-day albedo of trees and crops/grasses in CMIP5 all-forcings simulations.

17. As the Referee found the phrase "for which this information is available" line 464 redundant, we can reformulate the whole sentence.

18. As hinted by the Referee's comment, there was a mistake in the legend of Figure 1 and "absolute differences" should read "albedo values".

19. We have prepared scatter plots over common grid cells between subgrid and reconstructed estimates of albedo values and albedo differences between trees and crops/grasses in the CLM4.5 simulations. These plots will support the discussion in Section 3 of the performance of the first reconstruction method (described in Section 2.3.1).

20. The first reconstruction method (described in Section 2.3.1) requires information on the snow cover fraction (*snc*), which therefore limits the set of CMIP5 models that can be analysed compared to the second one (described in Section 2.3.2). Moreover, for the RF analysis we consider only the CMIP5 models for which at least two ensemble members are available in an attempt to limit the uncertainties of the method. These criteria overall explain that the Figures 5-10 and 11-13 are based on two different sets of models.

21. We had originally chosen not to present a model mean for Figure 12 because it would have required regridding the results from the individual models to a common grid, but this can be done for the revised manuscript.
* * *
[Figure]

the grid from a CMIP5 model
(original resolution)

the land grid cell
of interest *i*

the big box centered
over *i*

a snow-free grid cell
(*snc* < 0.1)

a non-land grid cell
(with land cover fraction < 50%,
excluded from the analysis)

a snow-covered grid cell
(*snc* > 0.9)

From all land grid cells within the big box:

$lcf_{tr}$, $lcf_{sh}$, $lcf_{cg}$, $snc$, $\alpha$

*at monthly resolution for 2000-2004*

$lcc_{tr->cg}$

*between pre-industrial and 1981-2000*
*(and $lon$, $lat$, $elev$)*

if at least 15 grid cells
are snow-free
(snc < 0.1) and
$lcf_{tr} + lcf_{sh} + lcf_{cg}$ > 90%

OR

if at least 15 grid cells are
snow-covered
(snc < 0.1) and
$lcf_{tr} + lcf_{sh} + lcf_{cg}$ > 90%

if at least 15 land grid
cells:

*Equations*
*(1), (3), (4)*

*Equations*
*(2), (5), (6)*

*Equations*
*(7), (8)*

Reconstructed albedo of
trees and crops/grasses in *i*
under snow-free conditions
$\alpha_{tr}^{sf}(i)$, $\alpha_{cg}^{sf}(i)$

Reconstructed albedo of trees
and crops/grasses in *i* under
snow-covered conditions
$\alpha_{tr}^{sc}(i)$, $\alpha_{cg}^{sc}(i)$

Reconstructed historical
albedo change associated with
conversions between trees
and crops/grasses in *i*

Figure 1: Description of the two employed reconstruction methodologies.
*snc* stands for snow cover fraction, $\alpha$ for albedo, *lcf* for land cover fraction,
*lcc* for land cover conversion, the suffixes *tr*, *sh* and *cg* for trees, shrubs and
crops/grasses, respectively, *lon* for longitude, *lat* for latitude, and *elev* for
elevation.

**Fig. 1.** new Figure to clarify the methodology

$$RF_{tr \leftrightarrow cg} = -\frac{1}{12}\sum\nolimits_{m=1}^{12} K_{\alpha_s,m}^{CACK} \times \Delta\alpha_{tr \leftrightarrow cg,m}$$

**Fig. 2.** Equation describing the calculation of the Radiative Forcing (see point 11 above)

---

## Author Response (AR1)

**General comments**

We thank the reviewers very much for their diligent review of our manuscript. Their comments have been very helpful to improve it. We insert below a point-by-point response to all of them.

The most important changes to be found in this revised version of the manuscript include:
- An update of the RF calculation method, which now makes use of the CACK v1.0 dataset
- The insertion of a Figure to visually explain the methodology of the reconstruction methods, as well as the reformulation of extensive parts of the Methodology section.
- The reconstruction of the present-day albedo of trees and crops/grasses for 3 more models (GFDL-CM3, GFDL-ESM2G and GFDL-ESM2M)
- The deletion of Table S1, its values being now mentioned in the text
- The addition of a supplementary table detailing which land cover classes from the GlobCover dataset corresponds to the broad classes employed in this study
- The addition of another supplementary table listing the CMIP5 models considered in the various analyses conducted for this study, as well their respective ensemble members
- References to previous studies on the topic pointed at by the reviewers

**Short Comment #1 from Ryan Bright**

This is an interesting and timely study. Although its novelty aspects pertain to the surface albedo extraction methods, the authors convert albedo changes to instantaneous radiative forcings (RF) which are subsequently benchmarked to results of several climate modeling studies and to IPCC AR5 estimates. Since the RF quantification and associated benchmarking is made an integral part of the paper, I would encourage the authors to reflect on the uncertainty of their RF estimates, which are based on a very simple parameterization [i.e., Eq. (12) and Cherubini et al. (2012)] that does not account for the spatio-temporal variation in atmospheric optical properties affecting transmittance of reflected solar radiation. Bright & O'Halloran (2019), for instance, benchmarked the performance of Eq. (12) to four GCM-based kernels and found persistent positive biases (see f. ex. Fig- ures 1 & 2) and a relative RMSE of about 20% globally. Bright & O'Halloran (2019) proposed a new simplified RF parameterization (see Eq. (17)) that sub- stantially reduces RF "error" (rRMSE of about 6% globally) and made a gridded RF kernel product based on this parameterization freely available (archived here: https://doi.org/10.6073/pasta/d77b84b11be99ed4d5376d77fe0043d8). This product is based on the same underlying CERES v4 data that has been employed in this study and includes uncertainty layers.

I would therefore encourage the authors to either: a) Drop the RF quantification and benchmarking part altogether and keep the focus on the novel albedo methods and merits, or b) Provide a strong justification for using Eq. (12) in light of its uncertainty, or c) Adopt an alternate RF kernel/model that has lower uncertainty.

Bright & O'Halloran 2019: https://www.geosci-model-dev.net/12/3975/2019/gmd-12-3975-2019.html

Thank you very much for the comment and the great suggestion. We have decided to use the CERES-based albedo change albedo kernel (CACK) from Bright and O'Halloran (2019) for the Radiative Forcing calculations. As a result, the values of global RF associated with

historical conversions between trees and crops/grasses are systematically less negative by ~20-30% for each of the CMIP5 models considered in Section 5 (see the revised versions of what are now Figures 12 and 13).

**Anonymous Referee #1**

Overall opinion: Lejeune et al. have devised an interesting and innovative method for extracting the albedo of forested and crop/grass land cover types from model simulations and the combination with the space-for-time approach to estimating the effect of land cover change is quite promising. I feel the science is of good quality and the results are useful.

Thanks to the reviewer for the useful and overall positive comments.

Some questions: I agree with the already posted comment that the RF estimates are based on a parameterization that may contribute its own biases and that the strength of the paper is in the novel albedo methods. I leave it up to the authors whether to address this in the discussion or to change the RF parameterization.

As specified in the response to Ryan Bright's comment, we have decided to follow his suggestion to use the CERES-based albedo change albedo kernel (CACK) from Bright and O'Halloran (2019) for the Radiative Forcing calculations. As a result, the values of global RF associated with historical conversions between trees and crops/grasses are systematically less negative by ~20-30% for each of the CMIP5 models considered in Section 5 (see the revised versions of what are now Figures 12 and 13).

Differences in soil type or texture can affect the albedo of vegetated surfaces and that this would add noise, if not bias to the 'space for time' method.

Surface albedo is indeed influenced by both the vegetation canopy and the soil, and this is the case in both satellite-derived observational products and climate models. It is true that both the vegetation canopy and the soil can exhibit variations in solar reflectance even for a same land cover, e.g. due to variations in Leaf Area Index or in soil texture. If such variations occur within a 'big box' of 5X5 grid cells, this can indeed introduce noise to the reconstruction methodology, and thus explain a substantial part of the RMSE of the reconstruction methodology discussed in Section 3.

Some brief discussion of the quality, accuracy or uncertainty in the datasets employed (e.g. ESA-CCI) would be helpful.

In the revised manuscript, we have added brief discussions of the quality of the employed observational datasets. The GlobAlbedo and MODIS MCD43C3 albedo products are considered to be of very good quality overall and show good agreement (global $R^2$ of 0.85). Some problems associated with snow detection were identified in GlobAlbedo but the resulting artifacts are most significant at very high latitudes (>70°), which are of lesser interest for our study (Muller et al, 2013). Imprecisions in land cover datasets such as GlobCover and ESA-CCI may especially arise via misclassification between land cover types within the broad trees or crops/grasses classes (e.g., between two types of trees) or the difficulty to properly identify medium-sized or mixed-type vegetation (i.e., shrub or savannalike). In contrast, because these products are best at distinguishing very distinct land cover types such as trees and crops/grasses, the satellite-derived albedo values of these two broad classes (retrieved following the methodology presented in Section 2.1.1) as well as their differences (obtained from the D18 data) are characterised by relatively low uncertainties.

I am curious as to why CRUNCEP V4 was chosen for offline simulations when newer versions and reportedly improved products such as CRUJRA and GSWP3 are available.

At the time the CLM4.5 simulation was conducted, CRUNCEP V4 was the recommended forcing data set (Chapter 26 in Oleson et al., 2013). Indeed, CLM offline simulations forced by GSWP3 represent surface albedo better than simulations forced by CRUNCEP (http://www.cesm.ucar.edu/models/cesm2/land/). However, the main purpose of this simulation is to demonstrate that the reconstruction method retrieves similar albedo alterations due to land-cover changes as the subgrid method. Therefore, the performance of this particular simulation is no major concern, as long as the simulated albedo is realistic to a sufficient extent.

Specific minor details:

Line 23: Doesn't constraining something usually reduce its range?

"Constraining" the global RF estimates by using the albedo changes due to conversions between trees and crops/grasses from satellite data instead of the individual CMIP5 models indeed leads to a somewhat unexpected increase in the model range. This result occurs because of two models which exhibit unrealistic historical conversion rates from trees to crops/grasses, which we therefore decided to discard for the final estimation of the RF from historical land-cover changes.

Line 26-28: Awk-ward sentence Multiple locations: "associated to" should be "associated with" Multiple locations: "inferior to" should be "less than" and "superior to" should be "greater than"

Line 300: Change "the local albedo difference between albedo and crops/grasses" to "the local albedo difference between forest and crops/grasses"

We have taken these remarks concerning the language into account when revising the manuscript and reformulated the relevant sentences.

References:

Muller, J.-P. et al. (2013) GlobAlbedo Final Product Validation Report.

Oleson, K. W. et al. (2013) 'NCAR/TN-503+STR NCAR Technical Note Technical Description of version 4.5 of the Community Land Model (CLM) Coordinating Lead Authors'. Available at: http://library.ucar.edu/research/publish-technote (Accessed: 31 December 2019).

**Anonymous Referee #2**

Summary: The authors attempt to train statistical models to extract the albedo of specific land cover classes in CMIP5 models, with the intent to then calculate the albedo change, and associated radiative forcing (RF), due to deforestation over the historical period. The paper is concise, and reasonably well written, although I found the description of the reconstruction methods to be somewhat unclear. The goals of the research are novel and highly relevant for the land surface and climate modelling communities, and I believe that this work will be suitable for publication once several important concerns are addressed.

We thank the reviewer for taking the time to go through the manuscript and submitting detailed comments. We are providing answers to these below, by referring to the in- dividual comments through mentions of the same line numbers given by the reviewer. We also attach a point-by-point response including the reviewer's comments, which may facilitate the second round of reviews.

Major comments:

-L158: I found the description of the reconstruction method hard to follow, perhaps because several different types of regression models were being applied simultaneously, and perhaps also because non-technical terms like "big box" were introduced. I think a simple diagram, showing the big box and the target (central) cell, and some of the most important quantities involved in the regression models, would be helpful to better explain the methods.

We have extensively worked on improving the methodology description. Including a figure was indeed a good idea to facilitate its understanding by the reader, and we have followed this piece of advice from the reviewer. The new Figure 1 should clarify non-common technical terms like "big box", the used reconstruction methods and some methodological steps we apply to increase their reliability.

-L235: The authors use an empirical parametrization to relate changes in surface albedo from deforestation to RF at Top of Atmosphere. While this approach is simple and straightforward, I wonder why the authors could not apply a surface albedo radiative kernel instead, as it removes the assumption of temporal and spatial homogeneity in atmospheric transmittance. Given that both surface and TOA clear sky kernels are publicly available, this minor methodological revision would be efficient and most useful. At the very least, the authors could validate their empirical parametrization against a radiative kernel with sample data. The authors are clear to cite the use of such a parametrization in other work. However, given that the the paper attempts to provide a precise, constrained RF estimate from historical deforestation, removing any limitation associated with such a result would provide a significant improvement to the manuscript.

All reviewers have suggested to use a different kernel to convert the reconstructed historical albedo changes due to conversions between trees and crops/grasses into RF estimates. Following Ryan Bright's comment, we have decided to use the version 1.0 of the CERES-based albedo change kernel (CACK) from Bright and O'Halloran (2019) for the Radiative Forcing calculations. This kernel is based on a novel, simplified parameterisation of shortwave radiative transfer and driven with downwelling shortwave radiation values at the surface and the top of the atmosphere obtained from the Clouds and the Earth's Radiant Energy System (CERES) Energy Balance and Filled (EBAF) 1°-resolution products. CACK

was evaluated by Bright and O'Halloran (2019): While being more easily understandable and easier to apply than kernels derived from climate models, it is able to mimick them more faithfully than five previously employed analytical, semi-empirical and empirical kernels.

-L284: In Figure 2, the authors show the reconstructed albedo of crops/grasses over northern Eurasia is essentially the albedo of snow (>0.8). Therefore, I take issue with the authors describing their reconstruction as the "extraction of the correct albedo values of specific land cover". This statement is true for the July reconstruction, but the underlying albedo of the vegetation in January is not 0.8, it is most likely very similar to the July value (~0.2). Therefore, to avoid any potential for confusing the reader, I would like the authors to describe clearly what is being extracted, which is the surface albedo of grid cells with different *underlying* land cover classes".

-L284: The reviewer is right that the surface albedo in both observational data and climate models is influenced by both the vegetation canopy and the soil reflectance. We have now clarified this at the beginning of Section 2.1.1. For the sake of simplicity, we however use the formulation "albedo of a specific land cover class" when referring to this mixed contribution of the soil and canopy to the surface albedo. This has also made clear in Section 2.1.1.

I note that the same issue also appears to be present in the reconstructed estimate of January albedos in Figures 6 and 8. Therefore, is it possible using this method to be certain that the albedo change due to LCC (e.g. Figure 10), and associated RF (Figure 11), is properly separated from the albedo change due to changes in snow accumulation and melt over the historical period?

We would like to stress that the present-day albedo of trees and crops/grasses is only reconstructed following the method described in Section 2.3.1 in order to be evaluated against satellite-derived data, as discussed in Section 4 and illustrated in Figures 6-11. In contrast, the historical albedo changes associated with transitions between trees and crops/grasses between the pre-industrial and 1981-2000 periods are reconstructed following the method described in Section 2.3.2, so that the associated global RF can be derived and discussed in Section 5 (based on Figures 12 and 13) in light of the model biases identified using the first reconstruction method. We acknowledge that this may have been ambiguous in the submitted manuscript, and intended to make it clearer in the revised version.

That being said, changes in surface albedo over vegetated surfaces between the pre-industrial and present-day periods are mainly influenced by changes in albedo of the vegetation canopy, in the fraction of ground (soil or snow) that is shed from sunlight by the vegetation canopy, and in the albedo of the ground. The first two contributions are mostly influenced by LCC and in particular transitions between trees and crops/grasses. They are therefore included in the term $\gamma_1$ of Equation (7). The latter contribution is mostly influenced by other climate forcings such as greenhouse gases, whose influence has a larger spatial extent which is thus assumed to be con-stant across a big box and included in the term $\gamma_0$. For the models for which factorial experiments (with LCC only or with all forcings except LCC) are available, we were able to directly extract the simulated change in surface albedo due to LCC. The similarities between these direct estimates and the results from the reconstructed method (compare the columns on the left and right sides of Figures S6, S7, S11 and S13) confirm the ability to properly separate surface albedo changes due to LCC from those due to changes in snow accumulation and melt. Some differences are found for the GFDL-ESM2 model but both estimates remain compatible given the uncertainty ranges of each method (see Figure S21).

-L311: In Section 3 the authors perform a validation of the reconstruction, and find errors in the reconstructed albedo in the range 10-40%. They conclude at the end of the Section that this method is appropriate to apply to CMIP5 models. But I would have appreciated a little more rigour in this part of the analysis; for example, the authors should define a priori what an acceptable tolerance of error would be. In other words, define what constitutes a "useful" estimate of albedo, which would provide the reader with a stronger basis for interpreting whether 10-40% error is acceptable. Since the focus later is on RF, perhaps one way to define "useful" is in terms of the perturbation that the uncertainty in albedo estimate passes on to the RF calculation, in energy units?

The present-day albedo of trees and crops/grasses that is reconstructed in the CMIP5 simulations using the methodology described in Section 2.3.1 is not used later on in the RF calculations, but simply to be evaluated against reference satellite-derived data. This evaluation effort reveals that, in some of the analysed CMIP5 models, the reconstructed albedo changes associated with transitions from trees to crops/grasses can differ from the reference values from Duveiller et al. (2018) by ~0.05 (respectively, ~0.4) over snow-free (respectively, snow-covered) areas (see Figures 10 and 11). These differences being substantially higher than the RMSE of the reconstruction (which amounts to ~0.02 over snow-free and ~0.05 over snow-covered areas, see Figures 4 and 5), we argue that the reconstruction method is useful to identify biases of CMIP5 models. We have included a paragraph justifying this thinking more extensively at the end of Section 3.2.

-L321: The authors find considerable intermodel differences in albedo biases, but I couldn't see any discussion linking the different biases to the underlying satellite-derived vegetation datasets used by each modelling group to calibrate their land models. In the case of CanESM2, it's the GLC2000, whereas for models using CLM it is MODIS. Could the authors investigate whether this difference is a contributing factor to the biases? And if so, perhaps the authors could make recommendations to the community as to which datasets produce the lowest biases?

Because our analysis focuses on the potential albedo change resulting from a land cover conversion between trees and crops/grasses rather than the mean surface albedo of each model grid cell, differences in the vegetation distributions of individual models should play a limited role in the biases identified in Figures 6-11. It is however true that if a model has a too low proportion of trees in a given region, for example, it can hinder the retrieval of the albedo of trees in this same region and therefore limits the scope of our analysis. However, we expect these effects to be of secondary importance.

L335: The discussion of biases in forested albedo when snow is present reminded me of the work by Thackeray et al. (2015, doi=10.1002/2015JD023325), and Wang et al. (2016, doi=10.1002/2015JD023824). I think that citations and connections to these previous studies would be helpful here to explain your results. In addition, Thackeray (2014, doi=10.1002/2014JD021858) shows that the parameterization of canopy albedo in CLM4 was overly sensitive to temperature, resulting in a seasonal cycle that differed significantly from observations.

The papers by Thackeray et al. (2014, doi=10.1002/2014JD021858 and 2015, doi=10.1002/2015JD023325), as well as Wang et al. (2016, doi=10.1002/2015JD023824) brought to our attention by the reviewer are indeed very relevant for the interpretation of our results. They point at model deficiencies which relate to some of the biases identified in our study, such as the too high albedo of trees in snow-covered areas in the MIROC5 and GFDL-ESM2M models. We have included references to these papers in the Results section (4.1.1), as well as in the Discussion and Conclusions section (6).

Minor comments:
-L82 and throughout: I suggest replacing all occurrences of "associated to" –> "associated with".
-L300: "albedo difference between albedo and crops/grasses" –> should this say "be- tween *trees* and crops/grasses"?
-L305: "remain similar" to what?
-L309: I suggest here citing some previous work on computing the "snow-masking effect of forests", for example by Essery (2013).

We have taken these minor comments into account when preparing the revised manuscript.

* * *
**Anonymous Referee #3**

This paper presents an approach to diagnosing CMIP5 model outputs with regard to the albedo changes and hence resultant radiative forcing from land cover change from trees to crops/grasses. It borrows the ideas from the analysis of observational data, that is, space for time to reconstruct albedo values of trees and crops/grasses and their differences via an unmixing technique based on linear regression over grid cells within local spatial windows. It compares results among CMIP5 models and between these models and observational data. The evaluation of the reconstruction approach using a model CLM also helps us understand the effectiveness of this approach that has been used to analyze observational data. The study uses both observational and modeling data that varies in terms of native spatial resolutions, temporal resolutions, and temporal coverages. This diversity in data strengths the investigation but also requires more efforts to achieve a clear and lucid description of methodology and results. I found the description of the methodology and the presentation of the results need some particular improvements. I admit that my knowledge is more on the observational side of studies on land cover and biophysical effects. Some of my questions may have common answers within the hard-core modelling community. Nonetheless, I believe addressing these issues would help the comprehension of the study by a wider audience.

Many thanks to the reviewer for taking the time to go through the manuscript and submitting such detailed comments. This is very helpful to improve the manuscript.
We are providing answers to the reviewer's comments below.

First, Section 2.3.1 needs some clarification in text. I have several questions concerning the understanding of the described method. See detailed comments below. In particular, the line 191-192 states a post-reconstruction estimates of albedo changes by calculating the differences in albedo between trees and crops/grasses. Then the section 2.3.2 looks like a

direct estimate of albedo changes from deforestation rate. So, what is the distinction between post-reconstruction estimates and the direct estimates in Section 2.3.2? And which estimates of albedo changes do you present in the results, e.g. Fig. 3, 4, and 9 to 13, and Table 1?

We indeed introduce two different reconstruction methods in the manuscript. We aim at clarifying the methodology section in a revised version of the manuscript, and especially at explaining better what these two methods intend to do and how. We also hope that the newly included Figure 1 will provide visual help in that respect. In this study we reconstruct two different quantities in CMIP5 models: 1) the simulated present-day albedo of trees and crops/grasses, to evaluate the albedo change arising from a potential transition between these two classes against observational data, and 2) the historical surface albedo changes associated with transitions between trees and crops/grasses, followed by an assessment of their consequence in terms of Radiative Forcing. Based on Figures 2-5, Section 3 focuses on the evaluation of the ability of the first employed reconstruction method to extract the first quantity (simulated present-day albedo of trees and crops/grasses) in CMIP5 all-forcings simulations. In Section 4, based on Figures 6-11 we evaluate the albedo change arising from a potential transition between trees and crops/grasses in CMIP5 models against observational data. This quantity has been extracted using the first reconstruction method, which has previously been evaluated in Section 3. In Section 5, based on Figures 12-14 we discuss the historical surface albedo changes associated with transitions between trees and crops/grasses between the pre-industrial and 1981-2000 periods, and which have been reconstructed in CMIP5 models using the second reconstruction method.

Second, as you use observational and modeling data of different temporal spans/coverages, please specify the year/temporal periods or the temporal coverages for all the figures and tables. For example, Fig 1 & 2, which year/temporal periods are you presenting? In particular for albedo changes, between which year/temporal periods are the presented difference, e.g., Fig. 3 & 4. Almost all the figure/table captions need such clarification. Also, better clarify the spatial resolution of your results. I haven't found explicit statement on the spatial resolution of your reconstructed albedo per land cover class or reconstructed albedo changes per land cover change from trees to crops/grasses. Line 158 says "big boxes of a size of 5 times 5 grid cells", and the grid cells of CMIP5 model outputs you are using have a size of 2 deg (line 90). Does this mean your reconstructed albedo values and albedo changes have a resolution of 10 deg?!

Following the reviewer's comment, we have now specified the relevant temporal periods in the legends of the Figures. Moreover, the revised methodology section as well as the new Figure 1 should make clearer that within one big box, we reconstruct the albedo values (or albedo changes) for the grid cell in the center of the big box, i.e. the reconstructed albedo values and albedo changes have the original model resolution (about 2°).

Third, about the radiative forcing from albedo changes. Here you are estimating and showing spatially-explicit RF. I'm not convinced that a single value for k (line 242-243) is enough to account for different solar angles at different latitudes. In Lenton and Vaughan (2009), they were looking into the global effect of geoengineering and a single value of k based on annual global mean transmittance was justified. But I'm convinced it is justified here. Also do you calculate RF from albedo changes month by month? Your presented results seems annual average of RF (fig. 11, 12). Then how do you average monthly changes since your reconstructed albedo changes clearly show monthly differences (fig. 3, 4, 9, 10)

All reviewers have suggested to use a different kernel to convert the reconstructed historical albedo changes due to conversions between trees and crops/grasses into RF estimates.

Following Ryan Bright's comment, we have decided to use the version 1.0 of the CERES-based albedo change kernel (CACK) from Bright and O'Halloran (2019) for the Radiative Forcing calculations. This kernel is based on a novel, simplified parameterisation of shortwave radiative transfer and driven with downwelling shortwave radiation values at the surface and the top of the atmosphere obtained from the Clouds and the Earth's Radiant Energy System (CERES) Energy Balance and Filled (EBAF) 1°-resolution products. CACK was evaluated by Bright and O'Halloran (2019): While being more easily understandable and easier to apply than kernels derived from climate models, it is able to mimick them more faithfully than five previously employed analytical, semi-empirical and empirical kernels. CACK provides a monthly climatology, thus since we reconstruct the albedo changes associated with historical conversions between trees and crops/grasses for each month (see Section 2.3.2), we can also compute an annual mean associated RF by averaging the contributions from each month (Equation 10).

Detailed comments,

1. Line 85-90, Please, even if just in supplementary materials, provide an explicit translation from GlobCover's land cover legend into your trees and crops/grasses for traceability.

As asked by the reviewer, we have included an explicit translation from GlobCover's land cover classification into the two broad classes that we used (trees and crops/grasses) in a new Table S1, and added a reference to this Table in Section 2.1.1.

2. Line 89, black sky albedo at what solar angle?

We understand that the GlobAlbedo product makes use of an optimal estimation approach including angular integrals and a gap-filling technique based on the MODIS surface anisotropy dataset in order to integrate data derived from the Advanced Along-Track Scanning Radiometer (AATSR), SPOT4-VEGETATION, SPOT5-VEGETATION2, and MERIS instruments, which exhibit different spectral and angular sampling. We have included this information in Section 2.1.1.

3. Section 2.1.2, specify the spatial resolution of D18 data in degrees for easier reading and easier comparison between presented datasets.

The resolution of the dataset from Duveiller et al. (1°) was missing and has now been specified in Section 2.1.2.

4. Line 131, Each grid cell in D18 dataset refers to a specific land cover change, i.e. a pair of land cover classes. What do you mean here by grouping land cover fractions in CMIP5 outputs within one land cover class? Furthermore, D18 provides albedo changes for 45 land cover transitions. How come the consistency with D18 is the reason for focusing just on transition between trees and crops/grasses?

As specified in Section 2.1.2, we have used the version of the D18 dataset that provides albedo changes for only six land cover transitions between four broad land cover classes (forests, shrubs, crops/grasses and savannas). This classification scheme is referred to as IGBPgen in the paper describing the dataset. Consistently with this scheme, when represented in the CMIP5 models we have considered grasses, crops and pasture as belonging to one single class: crops/grasses.

5.      Line 165, "inferior to" and "superior to". . . . maybe just simply say "less than".... ? simple words like "larger than" is enough and better for reading? There are more such cases in the rest of the text to be fixed.

We have now revised these formulations.

6.      Line 184 – 185, What does it mean by "land cover classes are represented" in a grid cell? lcf is larger than zero? If fewer than two grid cells have lcf > 0 for a class, then this class will not be considered in the regression at all. The Eq (2) will have one fewer term on the right side? But will that one grid cell with lcf > 0 (if there is one) for this class be used in such a regression that does not include this class? If so, isn't this inappropriate? If not, please clarify here.

The sentence originally containing "land cover classes are represented in a grid cell" has been reformulated as such: "Therefore, each predictor ($lcf_{tr}$, $lcf_{sh}$, $lcf_{cg}$ is only included in the regression (i.e., its corresponding term is included in Equation 1 or 2) if its value is greater than 0 in at least two snow-free (if $i$ is snow-free) or snow-covered grid cells (if $i$ is snow-covered)." If for example $lcf_{sh}$ is not included in a regression, that one grid cell with $lcf_{sh}$ > 0 (if there is one) may still be considered in the regression if over that grid cell $lcf_{tr}$ + $lcf_{cg}$ > 90%. This would effectively mean that $lcf_{sh}$ is less than 10%, therefore that the albedo of shrubs only accounts for a small portion of the mean surface albedo over that grid cell.

7.      Line 186–187,15 is more than half of 25 grid cells in a big box. So in each big box, you can only estimate albedo for either snow-covered land cover classes or snow-free land cover classes. But NOT for both snow-covered and snow-free?

The referee is correct that if a grid cell is snow-free (respectively, snow-covered) in a given month, we only estimate albedo for snow-free (respectively, snow-covered) conditions. We have reformulated parts of Section 2.3.1 to clarify this point, and the new Figure 1 should also help in that respect.

8.      Line 187, "where the sum of all the included predictors exceeds 90%.", Please clarify this sentence. Do you mean that at least 90% areal fraction of a big box, that is 0.9*25cells=22.5 cells-equivalent area, is needed for the sum of all the lcf over the included snow-covered OR snow-free grid cells? Or 90% is per EACH grid cell? This sentence reads like the former/first explanation.

We have reformulated the sentence originally present line 187 as such: "Moreover, the regressions are only conducted in the big boxes that have at least 15 grid cells (either snowfree or snow-covered) in which the sum of all the included predictors exceeds 90%." We hope that the inclusion of the new Figure 1 will also help to make this methodological point clearer.

9. Section 2.3.2, Eq. (9) and (10), Confusing symbols and texts here. What is difference between the meanings of the $\delta\alpha_{tr \to cg}$ in Eq. (9) and the $\delta\alpha_{tr \to cg}$ (i) in Eq. (10)? The $\delta\alpha_{tr \to cg}$ in Eq. (9) is albedo change as a results of transition from trees to crops/grasses. To me, this means the same as albedo changes due to deforestation that is defined by the $lcc_{tr \to cg}$, transition rate from trees to crops/grasses. If $\delta\alpha_{tr \to cg}$ in Eq (9) is a known quantity since you need it for the regression and it means the albedo change due to deforestation, what is the physical rational of the Eq (9)? And what is the point of using regression to achieve the estimate by Eq. (10)? And how did you get the value of $\delta\alpha_{tr \to cg}$ on the left side of the Eq. (9)? Is it from the methods given in Section 2.3.1 line 191-192? Please clarify.

There was indeed a mistake, there should have been no subscript on the term on the left side of Eq. 9 (now 7). This has now been corrected.

10. Line 218, "a jackknife resampling is conducted: Alternatively, and as", some typo here? esp. about the weird punctuation marks?

The description of the jackknife resampling at the end of Section 2.3.2 has been reformulated.

11. Line 248, the $\delta\alpha_{tr \to cg}$ is estimated per month. Do you also estimate RF per month?

We indeed estimate the instantaneous RF resulting from historical conversions from trees to crops/grasses for each month, then compute the annual mean value. We hope that the revised methodology description and in particular the newly introduced Equation 10 clarify this.

12. Line 250-251, it reads very unclear even considering the preceding texts and the context. Please elaborate on this.

In an attempt to clarify this Section 2.4, we now use different notations for the present-day potential surface albedo change associated with a transition from trees to crops/grasses ($\delta\alpha_{tr \to cg}$, reconstructed using the method described in Section 2.3.1) and the historical surface albedo changes due to conversions between trees and crops/grasses between the pre-industrial and 1981-2000 periods ($\Delta\alpha_{tr \to cg}$, reconstructed using the method described in Section 2.3.2). Newly introduced Equations (11) and (12) should also help clarify this.

13. Section 2.5, no information about how you estimated albedo changes of land cover change from trees to crops/grasses in this subgrid experiment. But your results presented such subgrid estimates of albedo changes (figure 3&4, table 1).

We have tried to formulate a clearer explanation of how the subgrid estimates of the present-day albedo of trees and crops/grasses are extracted from the CLM4.5 simulations.

Especially, the following sentences should help to understand this point: "Surface albedo values were output for each tile in these simulations, enabling to extract a subgrid albedo value for each land cover class (trees or crops/grasses, similarly as in Malyshev et al., 2015; Meier et al., 2018). For each grid cell and each month, the albedo values for these two land cover classes are computed as the area weighted mean albedo across each PFT pertaining to the respective class over the analysis period. This reference value, later referred to as "subgrid" estimate, can then be compared to the reconstructed albedo values."

Malyshev, S. et al. (2015) 'Contrasting Local versus Regional Effects of Land-Use-Change-Induced Heterogeneity on Histori- cal Climate: Analysis with the GFDL Earth System Model', Journal of Climate, 28, pp. 5448–5469. doi: 10.1175/JCLI-D-14-00586.1.

Meier, R. et al. (2018) 'Evaluating and improving the Community Land Model's sensitivity to land cover', Biogeosciences, 15, pp. 4731–4757. doi: 10.5194/bg-15-4731-2018.

14.    Line 263, what do you mean by "pixel" here? A grid cell in the CLM simulation? If so, be consistent in the terminology.

„Pixel" has been changed to „grid cell" in Section 2.5 to ensure consistency across the manuscript and more clarity for the reader.

15.    Line 263 – 265, Do you differentiate snow-covered and snow-free fractions of a grid cell when you calculate albedo values for trees or crops/grasses in this area-weighted average? If so, can you provide some explanation here?

We don't differentiate between snow-covered and snow-free grid cells when looking at the subgrid albedo values for trees and crops/grasses, but extract the subgrid albedo values for any snow cover fraction.

16.    Section 3.2, Which estimate of albedo changes are you presenting here, the estimate by Section 2.3.1, line 191-192, or the estimate by Section 2.3.2?

In Section 3 we focus on the evaluation of the ability of the first employed reconstruction method (presented in Section 2.3.1) to extract the simulated present-day albedo of trees and crops/grasses in CMIP5 all-forcings simulations.

17.    Line 464, what information in "for which this information is available"? This last part reads redundant and only adds confusion to this sentence.

The sentence has been reformulated: "We then demonstrated that the methodology gives estimates of the albedo of trees and crops/grasses that are close to the reference values provided at the sub-gridcell level in simulations for which these values available."

18.    Fig. 1 caption, what's the absolute difference here? You mean albedo values?

As hinted by the Referee's comment, there was a mistake in the legend of Figure 1 and "absolute differences" should read "albedo values". This has now been corrected.

19.     Almost all the figures of maps, good to have the maps here for spatial comparison. But it is only qualitative. Can you present a scatter plot over common grid cells between subgrid estimates and reconstructed estimates?

In Figures 2-5, we have now included scatter plots over common grid cells between subgrid and reconstructed estimates of albedo values (Figures 2&3) or albedo differences between trees and crops/grasses (Figures 4&5) in the CLM4.5 simulations. These plots support the discussion in Section 3 of the performance of the first reconstruction method (described in Section 2.3.1).

20.     Fig. 10 v.s. Fig. 11, why the different sets of models for albedo changes and RF?

20. The first reconstruction method (described in Section 2.3.1) requires information on the snow cover fraction (snc), which therefore limits the set of CMIP5 models that can be analysed compared to the second one (described in Section 2.3.2). Moreover, for the RF analysis we consider only the CMIP5 models for which at least two ensemble members are available in an attempt to limit the uncertainties of the method. These criteria explain why the Figures 6-11 and 12-14 are based on two different sets of models.

21.     Fig. 11, why not present a model mean as fig. 12?

21. We had originally chosen not to present a model mean for Figure 11 (now Figure 12) because it would have required regridding the results from the individual models to a common grid, but this is now done anyways to make the model results fit with the resolution of the CACK data. We have therefore also added a model mean for the unconstrained RF in the revised manuscript.

[revised manuscript text omitted]

Comment Text: Font: (Default) Times New Roman, Don't hyphenate

| Page 1: [2] Style Definition | 9/1/20 1:43:00 AM |
|---|---|

Footer: Suppress line numbers, Don't hyphenate

| Page 7: [3] Formatted | 9/1/20 1:42:00 AM |
|---|---|

Font: Times New Roman

| Page 7: [4] Formatted | 9/1/20 1:42:00 AM |
|---|---|

Font: Times New Roman

| Page 7: [5] Formatted | 9/1/20 1:42:00 AM |
|---|---|

Font: Times New Roman

| Page 7: [6] Formatted | 9/1/20 1:42:00 AM |
|---|---|

Font: Times New Roman

| Page 7: [7] Formatted | 9/1/20 1:42:00 AM |
|---|---|

Font: Times New Roman

| Page 7: [8] Formatted | 9/1/20 1:42:00 AM |
|---|---|

Font: Times New Roman

| Page 7: [9] Formatted | 9/1/20 1:42:00 AM |
|---|---|

Font: Times New Roman

| Page 7: [9] Formatted | 9/1/20 1:42:00 AM |
|---|---|

Font: Times New Roman

| Page 7: [10] Formatted | 9/1/20 1:42:00 AM |
|---|---|

Font: Times New Roman

| Page 7: [11] Formatted | 9/1/20 1:42:00 AM |
|---|---|

Font: Times New Roman

| Page 7: [12] Formatted | 9/1/20 1:42:00 AM |
|---|---|

Font: Times New Roman

| Page 7: [13] Deleted | 9/1/20 1:43:00 AM |
|---|---|

| Page 7: [14] Formatted | 9/1/20 1:42:00 AM |
|---|---|

Font: Times New Roman

| Page 7: [15] Formatted | 9/1/20 1:42:00 AM |
|---|---|

| Page 7: [17] Deleted | 9/1/20 1:43:00 AM |
|---|---|

| Page 7: [18] Formatted | 9/1/20 1:42:00 AM |
|---|---|

Font: Times New Roman

| Page 7: [18] Formatted | 9/1/20 1:42:00 AM |
|---|---|

Font: Times New Roman

| Page 7: [19] Formatted | 9/1/20 1:42:00 AM |
|---|---|

Font: Times New Roman

| Page 7: [20] Deleted | 9/1/20 1:43:00 AM |
|---|---|

| Page 7: [21] Formatted | 9/1/20 1:42:00 AM |
|---|---|

Font: Times New Roman

| Page 7: [22] Formatted | 9/1/20 1:42:00 AM |
|---|---|

Font: Times New Roman

| Page 7: [23] Formatted | 9/1/20 1:42:00 AM |
|---|---|

Font: Times New Roman

| Page 7: [24] Deleted | 9/1/20 1:43:00 AM |
|---|---|

| Page 7: [25] Formatted | 9/1/20 1:42:00 AM |
|---|---|

Font: Times New Roman

| Page 7: [26] Formatted | 9/1/20 1:42:00 AM |
|---|---|

Font: Times New Roman

| Page 7: [27] Formatted | 9/1/20 1:42:00 AM |
|---|---|

Font: Times New Roman

| Page 7: [28] Formatted | 9/1/20 1:42:00 AM |
|---|---|

Font: Times New Roman

| Page 7: [29] Formatted | 9/1/20 1:42:00 AM |
|---|---|

Font: Times New Roman

| Page 7: [30] Formatted | 9/1/20 1:42:00 AM |
|---|---|

Font: Times New Roman

| Page 7: [31] Formatted | 9/1/20 1:42:00 AM |
|---|---|

**Page 7: [34] Formatted**                                          9/1/20 1:42:00 AM

Font: Times New Roman

**Page 7: [35] Formatted**                                          9/1/20 1:42:00 AM

Font: Times New Roman

**Page 7: [36] Formatted**                                          9/1/20 1:42:00 AM

Font: Times New Roman

**Page 7: [37] Formatted**                                          9/1/20 1:42:00 AM

Font: Times New Roman

**Page 7: [38] Formatted**                                          9/1/20 1:42:00 AM

Font: Times New Roman

**Page 7: [39] Formatted**                                          9/1/20 1:42:00 AM

Font: Times New Roman

**Page 7: [40] Formatted**                                          9/1/20 1:42:00 AM

Font: Times New Roman

**Page 7: [41] Formatted**                                          9/1/20 1:42:00 AM

Font: Times New Roman

**Page 7: [42] Formatted**                                          9/1/20 1:42:00 AM

Font: Times New Roman

**Page 7: [43] Formatted**                                          9/1/20 1:42:00 AM

Font: Times New Roman

**Page 7: [44] Formatted**                                          9/1/20 1:42:00 AM

Font: Times New Roman

**Page 7: [45] Formatted**                                          9/1/20 1:42:00 AM

Font: Times New Roman

**Page 7: [46] Formatted**                                          9/1/20 1:42:00 AM

Font: Times New Roman

**Page 7: [47] Formatted**                                          9/1/20 1:42:00 AM

Font: Times New Roman

**Page 7: [48] Formatted**                                          9/1/20 1:42:00 AM

Font: Times New Roman

**Page 7: [49] Formatted**                                          9/1/20 1:42:00 AM

| Page 7: [52] Formatted | 9/1/20 1:42:00 AM |
|---|---|

Font: Times New Roman

| Page 7: [53] Formatted | 9/1/20 1:42:00 AM |
|---|---|

Font: Times New Roman

| Page 7: [54] Formatted | 9/1/20 1:42:00 AM |
|---|---|

Font: Times New Roman

| Page 7: [55] Formatted | 9/1/20 1:42:00 AM |
|---|---|

Font: Times New Roman

| Page 7: [56] Formatted | 9/1/20 1:42:00 AM |
|---|---|

Font: Times New Roman

| Page 7: [57] Formatted | 9/1/20 1:42:00 AM |
|---|---|

Font: Times New Roman

| Page 7: [58] Formatted | 9/1/20 1:42:00 AM |
|---|---|

Font: Times New Roman

| Page 7: [58] Formatted | 9/1/20 1:42:00 AM |
|---|---|

Font: Times New Roman

| Page 7: [59] Formatted | 9/1/20 1:42:00 AM |
|---|---|

Font: Times New Roman

| Page 7: [60] Formatted | 9/1/20 1:42:00 AM |
|---|---|

Font: Times New Roman

| Page 8: [61] Formatted | 9/1/20 1:42:00 AM |
|---|---|

Font: Times New Roman

| Page 8: [62] Formatted | 9/1/20 1:42:00 AM |
|---|---|

Font: Times New Roman

| Page 8: [63] Formatted | 9/1/20 1:42:00 AM |
|---|---|

Font: Times New Roman

| Page 8: [64] Formatted | 9/1/20 1:42:00 AM |
|---|---|

Font: Times New Roman

| Page 8: [65] Formatted | 9/1/20 1:42:00 AM |
|---|---|

Font: Times New Roman

| Page 8: [66] Formatted | 9/1/20 1:42:00 AM |
|---|---|

**Page 8: [69] Formatted**          **9/1/20 1:42:00 AM**

Font: Times New Roman

**Page 8: [69] Formatted**          **9/1/20 1:42:00 AM**

Font: Times New Roman

**Page 8: [70] Formatted**          **9/1/20 1:42:00 AM**

Font: Times New Roman

**Page 8: [71] Formatted**          **9/1/20 1:42:00 AM**

Font: Times New Roman

**Page 8: [72] Formatted**          **9/1/20 1:42:00 AM**

Font: Times New Roman

**Page 8: [73] Formatted**          **9/1/20 1:42:00 AM**

Font: Times New Roman

**Page 8: [73] Formatted**          **9/1/20 1:42:00 AM**

Font: Times New Roman

**Page 8: [74] Formatted**          **9/1/20 1:42:00 AM**

Font: Times New Roman

**Page 8: [75] Formatted**          **9/1/20 1:42:00 AM**

Font: Times New Roman

**Page 8: [76] Deleted**          **9/1/20 1:43:00 AM**

**Page 8: [77] Formatted**          **9/1/20 1:42:00 AM**

Font: Times New Roman

**Page 8: [78] Deleted**          **9/1/20 1:43:00 AM**

**Page 8: [79] Formatted**          **9/1/20 1:42:00 AM**

Font: Times New Roman

**Page 8: [80] Formatted**          **9/1/20 1:42:00 AM**

Font: Times New Roman

**Page 8: [81] Deleted**          **9/1/20 1:43:00 AM**

**Page 8: [85] Formatted**                                             **9/1/20 1:42:00 AM**

Outline numbered + Level: 3 + Numbering Style: 1, 2, 3, … + Start at: 1 + Alignment: Left + Aligned at:  0 cm + Indent at:  1,27 cm

**Page 8: [86] Formatted**                                             **9/1/20 1:42:00 AM**

Font: Times New Roman

**Page 8: [86] Formatted**                                             **9/1/20 1:42:00 AM**

Font: Times New Roman

**Page 8: [87] Formatted**                                             **9/1/20 1:42:00 AM**

Font: Times New Roman

**Page 8: [87] Formatted**                                             **9/1/20 1:42:00 AM**

Font: Times New Roman

**Page 8: [88] Formatted**                                             **9/1/20 1:42:00 AM**

Font: Times New Roman

**Page 8: [89] Formatted**                                             **9/1/20 1:42:00 AM**

Font: Times New Roman

**Page 8: [90] Formatted**                                             **9/1/20 1:42:00 AM**

Font: Times New Roman

**Page 8: [91] Formatted**                                             **9/1/20 1:42:00 AM**

Font: Times New Roman

**Page 8: [92] Formatted**                                             **9/1/20 1:42:00 AM**

Font: Times New Roman

**Page 8: [93] Formatted**                                             **9/1/20 1:42:00 AM**

Font: Times New Roman

**Page 8: [94] Formatted**                                             **9/1/20 1:42:00 AM**

Font: Times New Roman

**Page 8: [94] Formatted**                                             **9/1/20 1:42:00 AM**

Font: Times New Roman

**Page 8: [95] Deleted**                                             **9/1/20 1:43:00 AM**

▼

**Page 8: [95] Deleted**                                             **9/1/20 1:43:00 AM**

▼

| Page 8: [95] Deleted | 9/1/20 1:43:00 AM |
|---|---|

| Page 8: [96] Formatted | 9/1/20 1:42:00 AM |
|---|---|

Font: Times New Roman

| Page 8: [97] Formatted | 9/1/20 1:42:00 AM |
|---|---|

Font: Times New Roman

| Page 8: [98] Formatted | 9/1/20 1:42:00 AM |
|---|---|

Font: Times New Roman

| Page 8: [99] Formatted | 9/1/20 1:42:00 AM |
|---|---|

Font: Times New Roman

| Page 8: [100] Formatted | 9/1/20 1:42:00 AM |
|---|---|

Font: Times New Roman

| Page 8: [101] Formatted | 9/1/20 1:42:00 AM |
|---|---|

Font: Times New Roman

| Page 8: [102] Formatted | 9/1/20 1:42:00 AM |
|---|---|

Font: Times New Roman

| Page 8: [103] Formatted | 9/1/20 1:42:00 AM |
|---|---|

Font: Times New Roman

| Page 8: [104] Formatted | 9/1/20 1:42:00 AM |
|---|---|

Font: Times New Roman, English (UK)

| Page 8: [105] Deleted | 9/1/20 1:43:00 AM |
|---|---|

| Page 9: [106] Formatted | 9/1/20 1:42:00 AM |
|---|---|

Outline numbered + Level: 2 + Numbering Style: 1, 2, 3, … + Start at: 1 + Alignment: Left + Aligned at: 0 cm + Indent at: 0,74 cm

| Page 9: [107] Deleted | 9/1/20 1:43:00 AM |
|---|---|

| Page 10: [108] Formatted | 9/1/20 1:42:00 AM |
|---|---|

Outline numbered + Level: 1 + Numbering Style: 1, 2, 3, … + Start at: 1 + Alignment: Left + Aligned at: 0 cm + Indent at: 0,63 cm

| Page 10: [109] Formatted | 9/1/20 1:42:00 AM |
|---|---|

Outline numbered + Level: 2 + Numbering Style: 1, 2, 3, … + Start at: 1 + Alignment: Left + Aligned at: 0 cm + Indent at: 0,74 cm

**Page 28: [112] Deleted**                                              9/1/20 1:43:00 AM

**Page 30: [113] Deleted**                                              9/1/20 1:43:00 AM

**Page 30: [113] Deleted**                                              9/1/20 1:43:00 AM

**Page 31: [114] Deleted**                                              9/1/20 1:43:00 AM

**Page 33: [115] Deleted**                                              9/1/20 1:43:00 AM

**Page 33: [115] Deleted**                                              9/1/20 1:43:00 AM

**Page 33: [115] Deleted**                                              9/1/20 1:43:00 AM

**Page 33: [115] Deleted**                                              9/1/20 1:43:00 AM

**Page 33: [116] Deleted**                                              9/1/20 1:43:00 AM

**Page 33: [116] Deleted**                                              9/1/20 1:43:00 AM

**Page 33: [116] Deleted**                                              9/1/20 1:43:00 AM

---

## Author Response (AR2)

**General comments**

We would like to thank the reviewers for reading the revised manuscript. We have addressed the points raised by Referee #3, as described further below. Additionally, we would like to attract the reviewers' attention on a few other changes we had to introduce in this newly revised version compared to the last submitted one. We indeed came to realise that some of the publicly available CMIP5 data that is included in the presented analysis was incomplete. This affected the grass fraction for the HadGEM2-ES model (variable grassFrac). Based on a discussion with the relevant person at the modelling center that developed this model, we were able to reconstruct the grass fraction with another Tier-2 variable (landCoverFrac), which is also publicly available.

This issue overall doesn't not change the main conclusions of the paper. It however slightly affects its results in the following ways:
- Since we are now able to include the total amounts of grasses in our calculations for HadGEM2-ES, the present-day albedo response to conversions between trees and crops/grasses could be reconstructed for this model. The ensuing results are now included in Figures 6-9, and discussed in Section 4.
- The corrected historical conversion rates between trees and crops/grasses are more realistic, therefore the resulting Radiative Forcing from HadGEM2-ES is not an outlier any more, but close to the model mean. We are now therefore only excluding MIROC-ESM from the model subset to derive the model range, as it is the only remaining model with unrealistic historical conversion rates between trees and crops/grasses (see Section 5).

**Anonymous Referee #3**

We would like to thank again the Referee for their very diligent review of our manuscript and for providing relevant comments at such a level of detail.

* Line 205 – 209, $\alpha^{sf}$ or $\alpha^{sc}$ is estimated per month per year from 2000 to 2004 (according to line 164-165) at each grid cell, right? In Figure 6-9, "present-day" monthly albedo climatology is the average of monthly $\alpha^{sf}$ or $\alpha^{sc}$ of the years 2000 to 2004? If so, please specify it in the text and the figure caption (as in Figure for CLM, present-day (2002-2010)). Otherwise, please clarify.

Equation (1) describes the reconstruction of the monthly climatological values of the albedo of trees and crops/grasses. To conduct this reconstruction, we consider monthly climatological values of albedo and snow cover fraction over the 2000-2004 analysis period. A sentence has been added line 208 to clarify this. We have now also clarified the considered analysis periods in the legends of the Figures 6-11, as suggested by the Referee.

* Equation (10), Why do you divide the sum by 12? If $RF_{(tr \to cg)}$ is annual mean, shouldn't it be sum of monthly changes of all 12 months without being divided by 12? Otherwise, it would be a monthly mean? In contrast, in Equation (12), here you have a ratio of 1/12 because I believe $lcc_{(tr \to cg)}$ has a unit of area/year (though you did not clarify this in the text of section 2.3.2)? So 1/12 $lcc_{(tr \to cg)}$ would be monthly land cover conversion rate assuming equal conversion per month in a year. Then it multiplies with $\delta_{(tr \to cg,m)}^{D18}$ that is a monthly change in albedo per area?

We believe that dividing by 12 in Equation (10) is correct. The kernel is defined as the instantaneous change in outgoing shortwave radiation at the top of the atmosphere to an incremental change in surface albedo (thus being an energy flux, expressed in $W/m^2$). The CACK dataset provides monthly averages for this variable. We are interested here in the yearly average Radiative Forcing, thus the division by 12. In that regard, Equation (10) is also consistent with Equation (2) in Cherubini et al. (2012) or Equation (7) in Schwaab et al. (2015).

Cherubini et al., 2012. Site-specific global warming potentials of biogenic $CO_2$ for bioenergy: contributions from carbon fluxes and albedo dynamics. Environmental Research Letters

Schwaab et al., (2015). Carbon storage versus albedo change: radiative forcing of forest expansion in temperate mountainous regions of Switzerland. Biogeosciences

* Line 255, the γ coefficients are specific to each big box and ALSO each month?

That is correct, we have now specified this below the corresponding equation (line 261).

* Line 553, "have had little influence on the overall LCC-induced albedo changes", I think you meant to say "on the overall RF of LCC-induced albedo changes" or "on the overall LCC-induced RF"?

We argue that both statements are correct. The changes in background climate have had little influence on the overall RF of LCC-induced albedo changes because they have had little influence on these LCC-induced albedo changes in the first place. We have revised the sentence lines 559-564 to provide a more detailed reasoning.

* Figure 1, "if at least 15 grid cells are snow-covered (snc < 0.1) and", I think you meant "snc > 0.9" here? Also, you mean "at least 15 grid cells including the central grid cell i" to be exact, right?

That is right, we have now corrected this typo.

[revised manuscript text omitted]

**Page 2: [1] Formatted**                                             **10/16/20 2:41:00 PM**

English (UK)

**Page 2: [2] Change**                                     **Unknown**

Field Code Changed

**Page 2: [3] Formatted**                                             **10/16/20 2:41:00 PM**

English (UK)

**Page 2: [3] Formatted**                                             **10/16/20 2:41:00 PM**

English (UK)

**Page 2: [3] Formatted**                                             **10/16/20 2:41:00 PM**

English (UK)

**Page 2: [4] Formatted**                                             **10/16/20 2:41:00 PM**

English (UK)

**Page 2: [5] Change**                                       **Unknown**

Field Code Changed

**Page 2: [6] Formatted**                                             **10/16/20 2:41:00 PM**

English (UK)

**Page 2: [6] Formatted**                                             **10/16/20 2:41:00 PM**

English (UK)

**Page 2: [6] Formatted**                                             **10/16/20 2:41:00 PM**

English (UK)

**Page 2: [7] Change**                                       **Unknown**

Field Code Changed

**Page 2: [8] Formatted**                                             **10/16/20 2:41:00 PM**

English (UK)

**Page 2: [8] Formatted**                                             **10/16/20 2:41:00 PM**

English (UK)

**Page 2: [8] Formatted**                                             **10/16/20 2:41:00 PM**

English (UK)

**Page 2: [9] Change**                                       **Unknown**

Field Code Changed

**Page 2: [10] Formatted**                                           **10/16/20 2:41:00 PM**

**Page 2: [11] Change**                                         Unknown

Field Code Changed

**Page 2: [12] Formatted**                                       10/16/20 2:41:00 PM

English (UK)

**Page 2: [12] Formatted**                                       10/16/20 2:41:00 PM

English (UK)

**Page 2: [12] Formatted**                                       10/16/20 2:41:00 PM

English (UK)

**Page 2: [13] Change**                                         Unknown

Field Code Changed

**Page 2: [14] Formatted**                                       10/16/20 2:41:00 PM

English (UK)

**Page 2: [14] Formatted**                                       10/16/20 2:41:00 PM

English (UK)

**Page 2: [14] Formatted**                                       10/16/20 2:41:00 PM

English (UK)

**Page 2: [15] Change**                                         Unknown

Field Code Changed

**Page 2: [16] Formatted**                                       10/16/20 2:41:00 PM

English (UK)

**Page 2: [16] Formatted**                                       10/16/20 2:41:00 PM

English (UK)

**Page 2: [16] Formatted**                                       10/16/20 2:41:00 PM

English (UK)

**Page 2: [17] Change**                                         Unknown

Field Code Changed

**Page 2: [18] Formatted**                                       10/16/20 2:41:00 PM

English (UK)

**Page 2: [18] Formatted**                                       10/16/20 2:41:00 PM

English (UK)

**Page 2: [18] Formatted**                                       10/16/20 2:41:00 PM

**Page 2: [20] Formatted**          10/16/20 2:41:00 PM

English (UK)

**Page 2: [20] Formatted**          10/16/20 2:41:00 PM

English (UK)

**Page 2: [21] Change**          Unknown

Field Code Changed

**Page 2: [22] Formatted**          10/16/20 2:41:00 PM

English (UK)

**Page 2: [22] Formatted**          10/16/20 2:41:00 PM

English (UK)

**Page 2: [22] Formatted**          10/16/20 2:41:00 PM

English (UK)

**Page 2: [23] Change**          Unknown

Field Code Changed

**Page 2: [24] Formatted**          10/16/20 2:41:00 PM

English (UK)

**Page 2: [24] Formatted**          10/16/20 2:41:00 PM

English (UK)

**Page 2: [24] Formatted**          10/16/20 2:41:00 PM

English (UK)

**Page 2: [24] Formatted**          10/16/20 2:41:00 PM

English (UK)

**Page 2: [24] Formatted**          10/16/20 2:41:00 PM

English (UK)

**Page 2: [24] Formatted**          10/16/20 2:41:00 PM

English (UK)

**Page 2: [25] Formatted**          10/16/20 2:41:00 PM

English (UK)

**Page 2: [26] Change**          Unknown

Field Code Changed

**Page 2: [27] Formatted**          10/16/20 2:41:00 PM

**Page 2: [27] Formatted**                                                   **10/16/20 2:41:00 PM**

English (UK)

**Page 2: [28] Change**                                 **Unknown**

Field Code Changed

**Page 2: [29] Formatted**                                                   **10/16/20 2:41:00 PM**

English (UK)

**Page 2: [29] Formatted**                                                   **10/16/20 2:41:00 PM**

English (UK)

**Page 2: [29] Formatted**                                                   **10/16/20 2:41:00 PM**

English (UK)

**Page 2: [30] Change**                                 **Unknown**

Field Code Changed

**Page 2: [31] Formatted**                                                   **10/16/20 2:41:00 PM**

English (UK)

**Page 2: [31] Formatted**                                                   **10/16/20 2:41:00 PM**

English (UK)

**Page 2: [31] Formatted**                                                   **10/16/20 2:41:00 PM**

English (UK)

**Page 2: [31] Formatted**                                                   **10/16/20 2:41:00 PM**

English (UK)

**Page 2: [32] Formatted**                                                   **10/16/20 2:41:00 PM**

English (UK)

**Page 2: [33] Formatted**                                                   **10/16/20 2:41:00 PM**

English (UK)

**Page 2: [33] Formatted**                                                   **10/16/20 2:41:00 PM**

English (UK)

**Page 2: [34] Formatted**                                                   **10/16/20 2:41:00 PM**

English (UK)

**Page 2: [35] Change**                                 **Unknown**

Field Code Changed

**Page 2: [36] Formatted**                                                   **10/16/20 2:41:00 PM**